

# A two-dimensional Stockwell Transform for gravity wave analysis of AIRS measurements

N P Hindley[1], N D Smith[1], C J Wright[1], and N J Mitchell[1]

[1]Centre for Space, Atmosphere and Ocean Science, University of Bath, Claverton Down, Bath, UK

*Correspondence to:* Neil Hindley (n.hindley@bath.ac.uk)

**Abstract.** Gravity waves play a critical role in the dynamics of the middle atmosphere due to their ability to transport energy and momentum from their sources to great heights. The accurate parametrization of gravity wave momentum flux is of key importance to general circulation models. For the last decade, the nadir-viewing Atmospheric Infrared Sounder (AIRS) aboard NASA's Aqua satellite has made global, two-dimensional (2-D) measurements of stratospheric radiances in which
5 gravity waves can be detected. Current methods for gravity wave analysis of these data can introduce unwanted biases. Here, we present a new analysis method. Our method uses a 2-D Stockwell transform (2DST) to determine gravity wave horizontal wavelengths and directions in both directions simultaneously. We demonstrate that our method can accurately recover horizontal wavelengths and directions from a specified wave field. We show that the use of an elliptical spectral windowing function in the 2DST, in place of a Gaussian, can dramatically improve the recovery of wave amplitude. We measure momentum flux in
10 two granules of AIRS measurements in two regions known to be intense hot spots of gravity wave activity: (i) the Drake Passage/Antarctic Peninsula and (ii) the isolated mountainous island of South Georgia. We show that our 2DST method provides improved spatial localisation of key gravity wave properties over current methods. The added flexibility offered by alternative spectral windowing functions and scaling parameters presented here extend the usefulness of our 2DST method to other areas of geophysical data analysis.

## 1 Introduction

Gravity waves are a vital component of the atmospheric system. These propagating mesoscale disturbances can transport energy and momentum to great heights, where they are a key driving mechanism in the middle atmosphere through drag and diffusion processes (e.g. Fritts and Alexander, 2003, and references therein).

The accurate parametrization of unresolved gravity waves in global climate models (GCMs) has proven to be a long-standing
20 problem in the modelling community. The "cold pole" bias (Butchart et al., 2011), suffered by nearly all GCMs, has been suggested to be due to a deficiency of resolved and parametrized gravity wave drag near 60°S. One reason for this is that these gravity wave parametrizations continue to be poorly constrained by observations (Alexander et al., 2010). The accurate measurement of gravity wave properties is thus critical for the development of the next and current generation of climate models. In the last decade, satellite-based remote-sensing has greatly increased our capability to make gravity wave observations on a



global scale, but large discrepancies between observed and modelled fluxes still remain (Geller et al., 2013). This highlights the need for ever more accurate global and regional gravity wave measurements.

The Atmospheric Infrared Sounder (AIRS) (Aumann et al., 2003) is a nadir-sounding spectral imager on board the Aqua satellite, launched in 2002. Part of the A-Train satellite constellation, AIRS scans the atmosphere over the range $\pm 49°$ from the

nadir of the satellite in a 90 pixel ($\sim 1800\,\text{km}$) wide swath, using 2378 infrared channels along a sun-synchronous polar orbit. This continuous swath is archived in granules, usually 135 pixels ($\sim 2400\,\text{km}$) along-track. Stratospheric gravity waves can be detected in these granules as radiance perturbations in the 15 and 4.3 $\mu$m $CO_2$ emission bands (e.g. Alexander and Barnet, 2007; Hoffmann and Alexander, 2009; Hoffmann et al., 2013).

AIRS measurements enable the study of stratospheric gravity waves at unprecedented horizontal resolution. In order to fully

exploit these observations, accurate and easily-reproducible analysis methods for the measurement of gravity wave properties must be developed.

The Stockwell transform (S-transform) (Stockwell et al., 1996; Stockwell, 1999) is a widely-used spectral analysis technique for providing time-frequency (or distance-wavenumber) localisation of a time series (or profile). This capability makes the S-transform well suited to gravity wave analysis of a variety of geophysical data (e.g. Fritts et al., 1998; Alexander et al., 2008;

McDonald, 2012; Wright and Gille, 2013). The S-transform has also been used in a variety of other fields, such as planetary (Wright, 2012), engineering (Kuyuk, 2015) and medical sciences (Goodyear et al., 2004; Yan et al., 2015).

Alexander and Barnet (2007) developed a method for measuring gravity wave amplitudes, horizontal wavelengths and directions of propagation from AIRS granules using the one-dimensional S-transform. In their method, the S-transform is computed for each cross-track row, and co-spectra between adjacent cross-track rows are used to obtain spectral information in the along-

track dimension. To find the dominant waves in each granule, these cross-track co-spectra are averaged together and up to five peaks are located in each averaged spectrum. The method of Alexander and Barnet provides good first-order measurement of the properties of the (up to five) dominant wave features in a granule, but it can introduce unwanted biases as discussed further in Sect. 5.

Here, we present a new analysis method. AIRS radiance measurements are two-dimensional (2-D) images; thus a gravity

wave analysis method using a two-dimensional Stockwell transform (2DST) is a more logical approach. Here, we present a 2DST-based method for the measurement of gravity wave amplitudes, horizontal wavelengths, and directions of propagation from AIRS measurements. Our method takes advantage of the spatial-spectral localisation capabilities of the S-transform in both dimensions simultaneously, equally and without bias.

South Georgia and the Antarctic Peninsula, together with the southern tip of South America, lie in a well-known hot spot of

stratospheric gravity wave activity during austral winter, which has been extensively studied both observationally (Eckermann and Preusse, 1999; Alexander and Teitelbaum, 2007; Baumgaertner and McDonald, 2007; Hertzog et al., 2008; Alexander et al., 2009; Alexander and Teitelbaum, 2011; Alexander and Grimsdell, 2013; Hindley et al., 2015) and with numerical modelling techniques (Hertzog et al., 2008; Plougonven et al., 2010; Shutts and Vosper, 2011; Hertzog et al., 2012; Sato et al., 2012; Plougonven et al., 2013) in the last decade. These mountainous regions are subjected to a strong wintertime circumpolar flow

in the troposphere and stratosphere, and as a result are major orographic gravity wave sources (e.g. Hoffmann et al., 2013).



Despite this, discrepancies between observed and modelled gravity wave fluxes in this region are the largest anywhere on the planet (Geller et al., 2013). This unique geography of this important region provides a "natural laboratory" in which to make clear gravity wave measurements from space, and is thus an ideal region in which to test our 2DST methodology on AIRS measurements.

In Sect. 2 we introduce the 1-D and 2-D S-transform. In Sect. 3 we apply the 2DST to a specified wave field, describing our methodology for spatial localisation of the dominant spectral components. In Sect. 4 we present two alternative windowing functions for use in the 2DST. In Sect. 5 we apply the 2DST to two selected AIRS granules measured over the Antarctic Peninsula/Drake Passage and South Georgia. Finally, in Sect. 6 we summarise the key results of this study and discuss the advantages of our 2DST method in the context of previous work.

## 1.1 AIRS data

Gravity waves can be detected in AIRS radiance measurements as perturbations from a background state. Here, we use AIRS Level 1B radiance measurements from the 667.77 cm$^{-1}$ channel. These Level 1 radiances have considerably higher horizontal resolution than operational Level 2 temperature retrievals due to retrieval choices imposed on the latter (Hoffmann and Alexander, 2009). We compute brightness temperature $T$ directly from radiance $R$ as

$$\bar{T} = \frac{hc\nu}{k_B}\left(\ln\left(\frac{2hc^2\nu^3}{R}+1\right)\right)^{-1}, \tag{1}$$

where $h$ is Planck's constant, $c$ is the speed of light, $k_B$ is Boltzmann's constant and $\nu = 667.77$ cm$^{-1}$ is the wavenumber of the specified channel. Temperature perturbations $T'$ from the local background state $\bar{T}$ are then extracted via a fourth-order polynomial fit (Wu, 2004; Alexander and Barnet, 2007). This fit removes limb-brightening and other large-scale fluctuations. These brightness temperature perturbations are a more useful physical quantity with which to define gravity wave amplitudes, since gravity wave energies and momentum fluxes easier defined in terms of temperature perturbations (e.g. Ern et al., 2004).

The weighting function of the 667.77 cm$^{-1}$ channel peaks near 3 hPa ($\sim$40 km), with a full width at half maximum of $\sim$12 km (Alexander and Barnet, 2007, illustrated in Figure 1 of Wright et al. (2015a)). Gravity waves with vertical wavelengths shorter than 12 km are thus unlikely to be resolved and vertical wavelengths close to this limit will be strongly attenuated.

If the vertical wavelength is known, it is possible to correct for this attenuation by dividing the amplitude by an appropriate rescaling factor (Alexander and Barnet, 2007, their Fig. 4). Although methods for measuring long vertical wavelengths using multiple AIRS channels have been developed (e.g. Hoffmann and Alexander, 2009), we do not have direct measurements of vertical wavelength from our single AIRS channel, and so we do not apply such a correction to brightness temperature perturbations at this stage. The true amplitude of some waves in our initial analysis may therefore be between two and five times greater than the values shown. For the estimation of momentum flux in Sect. 5.3, we apply the attenuation correction described in Alexander and Barnet (2007).





## 2 The Stockwell transform

In its analytical form, the one-dimensional Stockwell transform (Stockwell et al., 1996) closely resembles a continuous wavelet

transform (CWT) with a complex sinusoidal mother wavelet windowed with a scalable Gaussian window (Gibson et al., 2006).

For time series data, this scalable Gaussian localises wave perturbations in the time domain through spectral localisation in the

frequency domain.

For a smoothly-varying, continuous and one-dimensional function of time $h(t)$, the generalised analytical form of the S-

transform $S(\tau, f)$ (e.g. Pinnegar and Mansinha, 2003) is given as

$$S(\tau, f) = \int_{-\infty}^{\infty} h(t)\omega(\tau - t, f)e^{-i2\pi ft}dt, \tag{2}$$

where $\tau$ is translation in the time domain, $f$ is frequency and $\omega(t - \tau, f)$ is a windowing function, scaled with frequency,

that provides spatial and spectral localisation. Traditionally, $\omega(\tau - t, f)$ takes the form of the normalised Gaussian window

$$\omega_{gau}(\tau - t, f) = \frac{|f|}{c\sqrt{2\pi}}e^{\frac{-(t-\tau)^2 f^2}{2c^2}} \tag{3}$$

where $c$ is a scaling parameter for the width of the Gaussian window, whose standard deviation $\sigma$ is scaled as $\sigma = \frac{c}{|f|}$. Thus

one has the familiar form of the S-transform

$$S(\tau, f) = \frac{|f|}{c\sqrt{2\pi}} \int_{-\infty}^{\infty} h(t)e^{-\frac{(t-\tau)^2 f^2}{2c^2}} e^{-i2\pi ft}dt, \tag{4}$$

The factor $c$ is a scaling parameter for the width of the Gaussian window (Mansinha et al., 1997a; Stockwell, 1999). Typically,

$c$ is set to 1 (e.g. Stockwell et al., 1996; Alexander et al., 2008; Wright and Gille, 2013), but may be set to other values to acheive

more specific time-frequency localisation requirements (e.g. Mansinha et al., 1997b; Fritts et al., 1998; Pinnegar and Mansinha,

2003). Setting $c > 1$ provides enhanced frequency localisation at the expense of time localisation, and contrarily setting $c < 1$

achieves enhanced time localisation at the expense of frequency localisation. We discuss this effect in more detail in Sect. 4.

To compute the S-transform using the form in Eq. 4, it seems we must compute the convolution of the Gaussian window and

the time series for each frequency voice $f$. However, considerable computational advantage can be achieved by rewriting the

S-transform as an operation in the frequency domain

$$S(\tau, f) = \int_{-\infty}^{\infty} H(\alpha)e^{-\frac{2\pi^2 c^2(\alpha - f)^2}{f^2}} e^{i2\pi\alpha\tau}d\alpha \tag{5}$$





where $\alpha$ is translation in the frequency domain and $H(\alpha)$ is the frequency analogue of $H(t)$. The Gaussian window $\omega(\alpha - f, f)$ now takes the form

$$\omega_{gau}(\alpha - f, f) = e^{\frac{-2\pi^2 c^2 (\alpha - f)^2}{f^2}} \tag{6}$$

where the standard deviation now scales with frequency as $\sigma = \frac{|f|}{2\pi c}$.

Here, the S-transform is computed for each frequency voice $f$ as the inverse Fourier transform of the product of $H(\alpha)$ and the corresponding frequency-domain Gaussian window $w(\alpha - f, f)$ in Eq. 6. By computing the S-transform via this frequency-domain multiplication, we avoid the computational expense of a convolution operation. Computationally efficient ("fast") discrete Fourier transform (DFT) algorithms are also used. The S-transform is most commonly implemented in this manner within the atmospheric sciences.

The S-transform has a number of desirable characteristics for geophysical data analysis. Unlike a CWT, the absolute magnitudes of the complex-valued S-transform coefficients in $S(\tau, f)$ may be directly interpreted as the instantaneous amplitude of the corresponding frequency voice $f$ at each location $\tau$. Information regarding wave amplitude is not strictly recoverable from a CWT, since the corresponding CWT coefficients are psuedo-correlation coefficients between the signal and the analysing wavelet.

One disadvantage to using DFT algorithms in an S-Transform implementation is the familiar coarse wavelength resolution at low frequencies, a limitation not encountered by the CWT. Since both the S-transform and DFT algorithms are easily extended to higher dimensions however, the reduced computational expense of a DFT-based S-transform makes this is a practical tool for large 2-D datasets. Retention of the wave amplitude information in the S-transform is another key advantage.

## 2.1   The two-dimensional Stockwell transform

The S-transform is easily extended to higher dimensions. For a two-dimensional 2-D image $h(x, y)$, the two-dimensional S-transform (2DST) is given by

$$S(\tau_x, \tau_y, k_x, k_y) = \frac{|k_x||k_y|}{2\pi c^2} \times \quad \ldots \tag{7}$$
$$\int_{-\infty}^{\infty} \int_{-\infty}^{\infty} h(x,y) e^{-\frac{(x-\tau_x)^2 k_x^2 + (y-\tau_y)^2 k_y^2}{2c^2}} e^{-i2\pi(k_x x + k_y y)} dx dy$$

where $\tau_x$, $\tau_y$, $k_x$ and $k_y$ are translation and wavenumber in the $x$ and $y$ directions respectively (Mansinha et al., 1997a;

Stockwell, 1999). Note that the transformed space has units of wavenumber $k_x, k_y$ rather than frequency, since we define an input image with units of distance. Here, the characteristic size of the Gaussian window is scaled as $\sigma_x = \frac{c}{|k_x|}$ and $\sigma_y = \frac{c}{|k_y|}$ in the same way as the one-dimensional form, where $c$ is the scaling parameter. The 2DST is introduced and well-described by Mansinha et al. (1997a) and Mansinha et al. (1997b), who demonstrated its promise for pattern analysis.





As with the one-dimensional S-transform, greater computational efficiency can be achieved by implementing the 2DST as an operation in the wavenumber domain as

$$S(\tau_x, \tau_y, k_x, k_y) = \int\limits_{-\infty}^{\infty} \int\limits_{-\infty}^{\infty} H(\alpha_x, \alpha_y) \times ...$$

$$e^{-2\pi^2 c^2 \left( \frac{(\alpha_x - k_x)^2}{k_x^2} + \frac{(\alpha_y - k_y)^2}{k_y^2} \right)} e^{i2\pi(\alpha_x \tau_x + \alpha_y \tau_y)} d\alpha_x d\alpha_y \qquad (8)$$

where $\alpha_x, \alpha_y$ are translations in the wavenumber domain and $H(\alpha_x, \alpha_y)$ is the wavenumber analogue of the input image $h(x, y)$.

The 2DST has been discussed and implemented in a variety of fields (Liu and Wong, 2007; Kocahan et al., 2008; Liu, 2009; Barry et al., 2012) but to our knowledge it has yet to be implemented in the atmospheric sciences. In the following section, we describe our 2DST implementation methodology for the purpose of gravity wave analysis from 2-D data.

## 3   Implementation of the 2DST on a specified wave field

To assess the capabilities of the 2DST, it is logical to first apply it to a two-dimensional specified wave field containing synthetic waves with known characteristics.

We create a specified wave field $h(x, y)$ with dimensions $100 \times 100$ km containing synthetic waves with unit amplitude and known wavelengths. Wave amplitudes are defined as temperature perturbations $T'$ in units of Kelvin. The synthetic waves are localised around their central locations with Gaussian functions (although note that they do overlap). We also add random ("salt and pepper") noise up to 10% of the wave amplitude.

We first compute the 2-D DFT $H(\alpha_x, \alpha_y)$ of our specified wave field $h(x, y)$. To recover an estimate of instantaneous wave amplitude, we use the familiar symmetry around the zeroth frequency in the Fourier domain to recover a 2-D analogy of the analytic signal. A 2-D DFT contains four quadrants that contain coefficients which are in complex-conjugate pairs with the coefficients in the opposite quadrant. The sum of these pairs always yields a real signal. By setting the coefficients of two of these quadrants to zero, and doubling their opposite quadrants, we obtain a complex-valued image when we take the inverse DFT. The magnitude of this image is analagous to the instantaneous amplitude, while the complex part descirbes instantaneous phase. All coefficients not in a complex conjugate pair are unchanged.

Next, we localise the wavenumber spectrum for each wavenumber voice $k_x$ and $k_y$ by multiplying $H(\alpha_x, \alpha_y)$ by the two-dimensional Gaussian window

$$\omega_{gau}(\alpha_x, \alpha_y, k_x, k_y) = e^{-2\pi^2 c^2 \left( \frac{(\alpha_x - k_x)^2}{k_x^2} + \frac{(\alpha_y - k_y)^2}{k_y^2} \right)}, \qquad (9)$$

whose widths in the $\alpha_x$ and $\alpha_y$ directions are scaled by wavenumbers $k_x$ and $k_y$ respectively and the scaling parameter $c$. We compute the 2DST $S(\tau_x, \tau_y, k_x, k_y)$ for each wavenumber voice $k_x$ and $k_y$ by sliding the Gaussian window through the wavenumber domain, taking the inverse 2-D DFT of each windowed spectrum.



For our purposes, we do not need to evaluate $S(\tau_x, \tau_y, k_x, k_y)$ for all positive and negative wavenumbers, since evaluating all positive and negative values of $k_y$ and only the positive values of $k_x$ gives us all the degrees of freedom. There is a residual $180°$ ambiguity in wave propagation direction which cannot be broken without additional information, which is supplied in Sect. 5.3.

A useful aspect of our implementation is that, like the CWT, we can compute the 2DST for any individual or range of permitted wavenumber voices by applying the appropriate wavenumber-scaled Gaussian windows. Although the permitted wavenumber voices in the spectral domain are evenly spaced, their corresponding wavelengths are limited to integer fractions of the number of elements in each dimension. This is an unavoidable consequence of using computationally-efficient DFT algorithms, which results in the familiar coarse spectral resolution seen at long wavelengths.

The ability to analyse an image at specific wavenumbers is a desirable aspect in geophysical data analysis, where some *a priori* information regarding the spectral range of wavenumbers detectable in a given dataset can be used to reduce the impact of unphysical, spurious or noisy results in 2DST analysis.

## 3.1 Measuring gravity wave properties

The 2DST $S(\tau_x, \tau_y, k_x, k_y)$ of our specified wave field is a four-dimensional object. For each location in $h(x, y)$, a two-dimensional complex-valued image of the localised spectral coefficients $\kappa(k_x, k_y)$ is evaluated.

Figure 1 shows a specified wave field $h(x, y)$ for which the 2DST has been computed. The absolute magnitude of the localised two-dimensional wavenumber spectrum $|\kappa(k_x, k_y)|$ is plotted for three different example locations. The coefficients of $|\kappa(k_x, k_y)|$ can be directly interpreted as the instantaneous amplitudes of waves with wavenumbers $k_x$ and $k_y$ at a given location in the specified wave field. As discussed in previous studies (e.g. Wright and Gille, 2013; Wright et al., 2015b), there are likely to be multiple peaks in $|\kappa(k_x, k_y)|$ corresponding to overlapping waves at the same location in $h(x, y)$. Indeed, in Fig. 1(b) we examine a location in the specified wave field where a small, high-wavenumber wave is located at the intersection of four lower-wavenumber waves. The localised spectrum computed by the 2DST shown in the foreground represents this feature well. The maximum spectral response is located in a peak at high $k_x$ and $k_y$ wavenumbers, with four smaller spectral peaks at lower wavenumbers with lower spectral responses.

A four-dimensional complex-valued function can be difficult to visualise. A more useful product might be a series of two-dimensional images, the same size as the input image, that contain the characteristics of the dominant wave at each location. In the implementation presented here, we neglect overlapping waves and identify a single dominant wave for each location in $h(x, y)$.

For each such location, we record the complex coefficient of $\kappa(k_x, k_y)$ located at the spectral peak of $|\kappa(k_x, k_y)|$. This yields one complex-valued image $\xi(\tau_x, \tau_y)$, with the same dimensions as the specified wave field $h(x, y)$, which contains the amplitude and phase of the dominant wave at each location.

The location of the spectral peak in $|\kappa(k_x, k_y)|$ also gives us the wavenumbers $k_x$ and $k_y$ to which this peak coefficient corresponds. Hence, we can produce two further images $K_x(\tau_x, \tau_y)$ and $K_y(\tau_x, \tau_y)$ which contain the dominant wavenumbers at each location in the specified wave field to which the coefficients of $\xi(\tau_x, \tau_y)$ correspond.





Thus, in the three images $\xi(\tau_x, \tau_y)$, $K_x(\tau_x, \tau_y)$ and $K_y(\tau_x, \tau_y)$, we can measure the amplitudes, phases, wavelengths and propagation directions of the dominant wave features at each location in our specified wave field.

Figure 2(a) shows our specified wave field $h(x, y)$. The central locations of the eight synthetic waves with unit amplitudes and known wavelengths are numbered 1 to 8.

By taking the real part of the complex-valued image $\xi(\tau_x, \tau_y)$ containing the dominant coefficients, we can recover a "reconstruction" of the specified wave field, and this is shown in Fig. 2(b). This is made possible by our analytic signal approach described above in Sect. 3.1.

The 2DST identifies the different spectral regimes of the specified wave field very well, but the reconstructed wave amplitudes are reduced by comparison to their original values.

We suspect the main reason for the reduced amplitudes relates to the "spreading" of spectral power in the transform. In practise, real-world waves are usually represented in the spectral domain as some combination of wavenumber voices, such that their spatial properties can be accurately recovered. This means that the spectral power of a single wave can be spread across multiple wavenumber voices. Spectral leakage contributes to this effect further.

The Gaussian window in the 2DST is equal to one at its central location, but immediately falls away with increasing radius.

This means that the spectral power that is contained in adjacent wavenumber voices, which is required to fully reconstruct the wave, is reduced. When the inverse DFT is computed, the recovered wave amplitude at this location is thus diminished.

Another reason for the diminished amplitude recovery in Fig. 2(b) is due to wave undersampling. This undersampling effect is worse for longer wavelengths, since fewer wave cycles are present in the same-sized region of the image. The wave undersampling limitations of the S-transform are well-understood in one dimension (Wright, 2010; Wright et al., 2015b).

Figure 2(c) shows the absolute magnitude of $\xi(\tau_x, \tau_y)$, which corresponds to the instantaneous amplitude of the dominant wave at each location. This output is useful for defining regions of the specified wave field that do or do not constrain the observed waves (McDonald, 2012).

The horizontal wavelength $\lambda_H(\tau_x, \tau_y) = (K_x^2 + K_y^2)^{-2}$ of the dominant wave at each location is shown in Fig. 2(d). Again, the different regimes of each wave in the specified wave field are clearly distinguished.

The direction of wave propagation $\theta(\tau_x, \tau_y)$, measured anticlockwise from the x-axis, is found as $\tan^{-1} \frac{K_x}{K_y}$, and plotted as Fig. 2(e). Note that $\theta(\tau_x, \tau_y)$ is subject to a $\pm\pi$ radian ambiguity, which is reconciled with *a priori* information in our AIRS analysis in Sect. 5.3.

To assess the effectiveness of our spectral analysis of the specified wave field, we compare the known wavelengths and propagation angles of the synthetic waves in the test image with the 2DST-measured wavelengths and propagation angles in

Fig. 2(f). For each wave numbered 1 to 8, blue dots show the input wavelength $\lambda_{IN}$ against measured wavelength $\lambda_{OUT}$, indicating that the 2DST measures the horizontal wavelengths and propagation angles in the test image very well. Generally, shorter wavelengths are well-resolved but longer wavelengths are slightly underestimated. This is likely due to the coarse spectral resolution of DFT-based methods for waves with wavelengths that are a large fraction of the image size, since such waves are more susceptible to spectral leakage problems.





## 4 Alternative spectral windowing methods

The use of the Gaussian window in the S-transform has some convenient mathematical advantages; it is analytically simple and has a definite integral over an infinite range. However, an unfortunate side effect in two-dimensions is the poor recovery of wave amplitude. Although a Gaussian is traditionally used, any suitable apodizing function may be used, as long as its spatial

integral is equal to unity (e.g. Stockwell, 2007). For example, Pinnegar and Mansinha (2003) used an asymmetric hyperbolic time-domain window for enhanced measurement of the onset times of one-dimensional time series components.

In this section, we investigate the use of two alternative two-dimensional windowing functions in the 2DST. We first consider an elliptical spectral window $\omega_{ell}$, which corresponds to a sinc-function-shaped window in the spatial domain. We then consider a sinc function spectral window $\omega_{sinc}$, which corresponds to a pseudo-elliptical-shaped window in the spatial domain. These

two functions represent logical extremes for apodizing functions in the 2DST.

We find that the use of an elliptical windowing function in the spectral domain can greatly improve amplitude recovery while maintaining good spatial and spectral resolution.

### 4.1 Elliptical window

As discussed above, the spectral peaks in a DFT spectrum have a characteristic width, where the spectral power is spread in a

broad peak around the central wavenumber. This spectral power is slightly reduced when a Gaussian window is applied, due to the immediate decrease in the Gaussian function around the central location. This effect is illustrated for a one-dimensional Gaussian window (solid red line) in Fig. 3(a). The effect can be mitigated, but not fully reconciled, by decreasing the scaling parameter $c$, which broadens the spectral Gaussian window. However, this decreases the width of the spatial window, which increases the effect of wave undersampling for low wavenumbers.

One solution to this problem is to define an elliptical windowing function $\omega_{ell}(\alpha_x, \alpha_y, k_x, k_y)$ in the spectral domain as

$$\omega_{ell}(\alpha_x,\alpha_y,k_x,k_y) = \begin{cases} 0 & \text{for} & 2\pi c\sqrt{\frac{(\alpha_x-k_x)^2}{k_x^2} + \frac{(\alpha_y-k_y)^2}{k_y^2}} & \geqslant & 1 \\ 1 & \text{for} & 2\pi c\sqrt{\frac{(\alpha_x-k_x)^2}{k_x^2} + \frac{(\alpha_y-k_y)^2}{k_y^2}} & < & 1 \end{cases} . \tag{10}$$

The exact width of this elliptical window is scaled in the $\alpha_x$ and $\alpha_y$ directions as twice the standard deviation of the equivalent Gaussian window in Eq. 9. This is illustrated in Fig. 3(c). This elliptical window captures the full extent of the targeted spectral peak at $k_x$ and $k_y$, greatly improving wave amplitude recovery. The width of $\omega_{ell}$ can also be more carefully adjusted using the

scaling parameter $c$. This elliptical window can then be used in place of the Gaussian windowing term in Eq. 8.

The use of an elliptical window in the 2DST significantly improves wave amplitude recovery when compared to the equivalent Gaussian (see Fig. 4). This is very useful in our analysis of AIRS data in Sect. 5.



## 4.2 Sinc window

Another windowing approach is to use a scalable sinc function $\omega_{sinc}(\alpha_x, \alpha_y, k_x, k_y)$ for spectral domain localisation, illustrated in Fig. 3d. If we define our sinc window as

$$\omega_{sinc}(\alpha_x, \alpha_y, k_x, k_y) = \frac{\sin\left(2\pi c\sqrt{R_{\alpha_x}^2/k_x^2 + R_{\alpha_y}^2/k_y^2}\right)}{2\pi c\sqrt{R_{\alpha_x}^2/k_x^2 + R_{\alpha_y}^2/k_y^2}} \tag{11}$$

where $R_{\alpha_x} = (\alpha_x - k_x)$ and $R_{\alpha_y} = (\alpha_y - k_y)$, we see that in the spatial domain this closely resembles an elliptical window of instantaneous radius $R_{x,y}^2 = (c/k_x)^2 + (c/k_y)^2$. Thus, for all real positive values of $c$, this sinc window has the same width in the spatial domain as twice the standard deviation of the original Gaussian window defined in Eq. 8. Though some artefacts may persist due to edge truncation function at high wavenumbers, such a sinc function can provide a strict elliptical localising window in the spatial domain with an abrupt edge, giving improved distinction between different wave regimes in the input

image using the 2DST. As with both the elliptical and Gaussian windowing functions, the characteristic width of the sinc window can be adjusted using the scaling parameter $c$.

## 4.3 The effect of window choices on AIRS granules

Figure 4 shows an AIRS granule over the Southern Andes measured on 24[th] May 2008, analysed using the 2DST with three different windowing approaches.

In Figs. 4(b) and 4(e), we use a Gaussian windowing function with the scaling parameter $c$ equal to one. This is the window usually used in 1DST implementations. We see that, as discussed above, this choice of window is only able to recover the very general, long-horizontal wavelength features of the granule, with poor spatial localisation and significantly reduced amplitude. This is due to a large proportion of the spectral response being lost by the windowing Gaussian when applied to two dimensions.

We can reduce the impact of this by decreasing the scaling parameter $c$, which broadens (narrows) the spectral (spatial) win-

dow. This provides improved amplitude recovery and improved spatial localisation at the expense of spectral localisation (Fritts et al., 1998). Since we only select a single dominant spectral peak for each location on the granule, this is acceptable for our purposes. The "reconstructed" perturbations and horizontal wavelengths (Figs. 4(c) and (f)) are now much more representative of the wave features in the granule.

One problem remains, however. By decreasing $c$, we narrow our spatial window. In regions where wave amplitudes are low,

such as the bottom-left corner of Fig. 4(a), this narrow Gaussian window starts to undersample long wavelengths, such that only very short wavelengths are attributed to the region. The elliptical window used in Figs. 4(d) and (g) performs better at recovering the underlying larger-scale structure of the granule, without defaulting to the small-scale noisy variations due to undersampling. Amplitude recovery at all wavelengths is also improved over either of the Gaussian approaches.

In the general case, these low-amplitude, small-scale variations are unlikely to be due to gravity waves with vertical wave-

lengths visible to AIRS, so their recovery is something we try to avoid. Furthermore, such wavelengths are very close to or at




the Nyquist limit for these data. Our confidence in their measurement is thus very low, yet the momentum fluxes they transport can dominate. We discuss this further in Sect. 5.4.

For the windowing functions considered, it is clear from Fig. 4 that the scaling parameter $c$ has a first-order effect in determining the spatial-spectral localisation capabilities of the 2DST. The elliptical windowing function, with a scaling parameter of $c = 0.25$, was selected for our AIRS analysis in the next section. This choice provided the best trade-off between spatial and spectral localisation of different wave regimes in AIRS measurements. The use of a sinc window, (omitted for brevity) provides spatial-spectral localisation results roughly halfway between the Gaussian and elliptical windows shown in Fig. 4 when the same scaling parameter value is used.

## 4.4 Invertibility

A very convenient aspect of the S-transform is its invertibility, namely that the temporal (or spatial) sum of the S-transform $S(\tau, f)$ is equal to the Fourier transform $H(f)$:

$$\int_{-\infty}^{\infty} S(\tau, f) \, d\tau = H(f), \tag{12}$$

which can be easily inverted to recover the original signal. This feature of the S-transform is dependent on the requirement that the temporal (or spatial) sum of the selected windowing function $\omega$ is equal to unity, namely

$$\int_{-\infty}^{\infty} \omega(\tau - t, f) \, d\tau = 1, \quad \text{or} \quad \int_{-\infty}^{\infty} \omega(\tau_x - x, k_x) \, dx = 1. \tag{13}$$

By taking the spatial sum of the Gaussian window in Eq. 4, we see that the normalisation term $\frac{|k_x||k_y|}{2\pi c^2}$ ensures that this condition is satisfied. The spatial domain form of our spectral-domain elliptical windowing function $\omega_{ell}(\alpha_x, \alpha_y, k_x, k_y)$ in Eq. 10 is sinc-shaped and given by

$$\omega_{ell}(\tau_x, \tau_y, k_x, k_y) = K_{ell} \times \text{sinc}\left(\frac{1}{2\pi c}\sqrt{(x - \tau_x)^2 k_x^2 + (y - \tau_y)^2 k_y^2}\right) \tag{14}$$

where $K_{ell}$ is a normalisation factor and the sinc function is the unnormalised form (no factor of $\pi$). Unfortunately, the spatial integral of the sinc term in 14 does not have a definite value, which means we cannot obtain an exact expression for $K_{ell}$. However, a good approximation can be made if we take $K_{ell} = \frac{|k_x||k_y|}{4\pi c^2}$. This is equal to one half of the normalisation term used for the Gaussian window. Using this approximation, we are able to invert the 2DST when our elliptical windowing function is used, although this approximation can lead to under-representation of very low wavenumbers.

This means that, if required, the 2DST can be exactly inverted for the Gaussian and pseudo-inverted for the elliptical windowing functions presented here. Thus, the original 2-D image can be recovered from the inverse DFT of the spatial sum of the 2DST. Note that a traditional CWT does not have this capability. The fact that we can achieve such flexibility in spatial-spectral resolutions by swapping our Gaussian window for an elliptical or by adjusting $c$, yet still retain the capability of inversion, further highlights the strength of the 2DST as a tool for spatial-spectral analysis of geophysical data.





Unfortunately, to take full advantage of DFT algorithms and the inversion capability of the 2DST for AIRS data, we must compute the 2DST using all permitted wavenumber voices in both dimensions. This requires nearly 12 000 inverse DFT calculations for each AIRS granule using the traditional voice-by-voice implementation described here. Interpolating AIRS measurements to a coarser resolution with fewer pixels could be one solution, but this will obviously undersample short horizontal

wavelengths in the data. Faster methods for computing the S-transform have been developed (Brown et al., 2010) which may increase practicality in the furture. Other steps, such as avoiding loops and ensuring that 2-D objects to be transformed have dimensions that are powers of two, may also reduce computational expense.

## 5    AIRS gravity wave analysis using the 2DST

In this section, we use our 2DST-based method to perform gravity wave analysis on two-dimensional granules of AIRS radiance

measurements, comparing our analysis to that of previous studies. We use the 2DST to measure gravity wave amplitudes, horizontal wavelengths, and directions of propagation. We then use ECMWF-derived wind speeds and the assumption of an orographic wave source to infer vertical wavelengths and make estimates of gravity wave momentum flux (the vertical flux of horizontal pseudomomentum) by closely following the method of Alexander et al. (2009).

### 5.1    AIRS granule selection and pre-processing

The first AIRS granule selected for our study is granule 32 of 6$^{th}$ September 2003, over South Georgia. The second granule is a 135-pixel swath over the intersection between granules 39 and 40 on 2$^{nd}$ August 2010, located over the Antarctic Peninsula and Drake Passage.

     Alexander et al. (2009) and Hoffmann et al. (2014) performed an analysis of these AIRS granules over South Georgia and the Antarctic Peninsula respectively. Their studies measured wave amplitudes, horizontal wavelengths and wave propagation

directions using a one-dimensional S-transform method, as described by Alexander and Barnet (2007). In their method, the one-dimensional S-transform is computed for each cross-track row. Then, covariance spectra are computed between pairs of adjacent cross-track rows to measure phase shifts in the along-track direction, from which along-track wavelengths can be inferred. To find the dominant wave features in a granule, these co-spectra are averaged together and up to five spectral peaks are found in this averaged spectrum. This approach can provide computationally fast, first-order gravity wave analysis of AIRS

granules, and it has been used in numerous other studies (e.g. Alexander and Teitelbaum, 2011; Alexander and Grimsdell, 2013; Wright et al., 2015a).

     One limitation of this method is that the phase difference measurements required to recover along-track wavenumbers can introduce a strong cross-track bias in resolved features, since the S-transform is only computed in the cross-track direction. In addition, waves which occupy only small regions of the granule in the along-track direction may also be under-represented in

the averaged co-spectrum. Furthermore, selecting no more than five dominant waves in the averaged co-spectrum implicitly limits the maximum number of available along-track wavenumber voices to no more than five for each location on the entire granule. The use of a two-dimensional Stockwell transform is a logical solution to each of these problems. With the increased





convenience of computational power since the study of Alexander and Barnet (2007), the 2DST now represents a more practical alternative to the one-dimensional method.

Before implementing the 2DST, each granule of brightness temperature perturbations is interpolated onto a regularly-spaced grid with approximately 17.7 km and 20.3 km separating adjacent pixels in the along-track and cross-track directions respec-

tively.

As a result of using computationally-efficient two-dimensional DFT algorithms, the maximum numbers of permitted wavenumber voices available in the along-track and cross-track directions are limited to $N_{AT} - 1$ and $N_{XT} - 1$, where $N_{AT}$ and $N_{XT}$ are the number of pixels in the along-track and cross-track directions (135 and 90 respectively). These wavenumber voices have corresponding wavelengths that are integer fractions of the total along-track and cross-track dimensions of the granule.

Here, we compute the 2DST for wavelengths greater than around 40 km. This is just over twice the Nyquist-sampling distance between AIRS pixels after interpolation onto our regular grid. The zeroth frequency (the spatial mean of the granule) is omitted. Increased along-track spectral resolution at low wavenumbers can be obtained by applying the 2DST to two or more adjacent granules, thus increasing the number of along-track pixels.

## 5.2   AIRS gravity wave properties measured by the 2DST

The results of our 2DST analysis of the selected AIRS granules over South Georgia and the Antarctic Peninsula are shown in Figs. 5 and 6 respectively.

In both Figures, panel (a) shows the brightness temperature perturbation measurements calculated as described in Sect. 1.1. Note that the colour scale is chosen so as to make wave perturbations clearer by eye, but at some locations it is saturated.

Clear wave-like perturbations are observed in both granules directly over and to the east of the mountain ranges. As in

previous work, such clear wave-like perturbations are attributable to gravity waves with a high degree of certainty.

Reconstructed 2DST temperature perturbations $T'_{2DST}$ are shown in panel (b). These are found by taking the real part of the complex 2DST object $\xi(\tau_x, \tau_y)$ as described in Sect. 3.1.

The image $T'_{2DST}$ shows the dominant wave features in the granule reconstructed using only the pre-defined range of permitted wavenumber voices in the 2DST. Since we only consider the coefficients of the dominant wavenumber at each

location, this reconstruction cannot be perfect, but it provides a visual inspection of how well the 2DST outputs represent the dominant wave characteristics of the granule. The $T'_{2DST}$ image can be used to "fine-tune" the 2DST by changing the windowing function, by adjusting the scaling parameter $c$, or by redefining the range of frequency voices until the desired outcome is achieved. Such fine-tuning flexibility cannot be so easily achieved using the 1DST method.

Panels 5(c) and 6(c) show instantaneous wave amplitude $|T'|_{2DST}$ for each granule. This is found by taking the absolute

magnitude of the complex 2DST object $\xi(\tau_x, \tau_y)$ as described in Sect. 3.1. This property provides us with a useful metric with which to define regions of the granule which do or do not contain wave-like perturbations, such that we can limit spurious detections (e.g. McDonald, 2012). In Figs. 5(h) and 6(c–h), we exclude regions of each granule where the instantaneous wave amplitude is more than one standard deviation below the mean of the wave amplitudes. In Figs. 5(c–g), we do not exclude such regions for discussion purposes, so as to provide an example of the data we would otherwise omit.





Panels 5(d) and 6(d) show absolute horizontal wavelengths $\lambda_H = (k_{AT}^2 + k_{XT}^2)^{-1/2}$, where $k_{AT}$ and $k_{XT}$ are the along-track and cross-track wavenumbers respectively. We can see that these horizontal wavelengths clearly define different regimes of the dominant wave features of the granules, as in the test case in Sect. 3.1, though the AIRS data are more complex. In the South Georgia granule in Fig. 5(d), we see that the island lies within a wave field where long horizontal wavelengths are dominant

around and to the east of the island over the ocean, with their wavenumber vectors aligned roughly parallel to the direction of the mean flow. This is characteristic of a wing-shaped mountain wave field (Alexander and Grimsdell, 2013), and is in good agreement with visual inspection of the granule itself.

In panels 5(e) and 6(e), we show the orientation of the horizontal wavenumber vector measured anticlockwise from east. $\theta$ is calculated by first projecting the along-track and cross-track wavenumber vectors $k_x$ and $k_y$ into their zonal and meridional

components $k$ and $l$ using the azimuths of the along-track and cross-track directions at each location on the granule, then taking $\theta = \tan^{-1}(\frac{l}{k})$. Note that $\theta$ only describes the orientation and not the true horizontal direction of propagation of the wavenumber vectors, which retain a $\pm 180°$ ambiguity that we break below.

In the South Georgia granule (Fig. 5), we see that our 2DST measurements in the southern portion of the granule are largely dominated by small-scale, low-amplitude, short horizontal wavelength features with random directions of propagation. Most of

these features are likely to be due to noise and not attributable to coherent wave structures. By using a threshold amplitude, such regions are effectively removed, leaving well-defined regions with clear wavelike perturbations. The contribution of small-scale features than remain after this step is discussed further in Sect. 5.4.

## 5.3   Momentum fluxes

Here we make estimates of gravity wave momentum flux for the dominant wave-like features measured by the 2DST in our

selected granules, following the method of Alexander et al. (2009).

The zonal and meridional components of gravity wave momentum flux $\mathrm{MF}_x$ and $\mathrm{MF}_y$ are given by

$$\mathrm{MF}_{x,y} = \frac{\rho}{2} \left( \frac{g}{N} \right)^2 \left( \frac{T_a'}{\bar{T}} \right)^2 \left( \frac{k}{m}, \frac{l}{m} \right) \tag{15}$$

where $\rho$ is density at a height of $40\,\mathrm{km}$, $g$ is the acceleration due to gravity, $N$ is the buoyancy frequency, $T_a'$ is the attenuation-scaled instantaneous wave amplitude, $\bar{T}$ is the background temperature, and $k$, $l$ and $m$ are wavenumbers in the zonal, merid-

ional and vertical directions respectively (Ern et al., 2004).

We have a $\pm 180°$ ambiguity in direction of propagation, which we break by assuming the waves in our granules always propagate against the mean flow. The last unknown in Eq. 15 is vertical wavenumber $m = 2\pi/\lambda_Z$. Under the assumption that the waves are upwardly propagating mountain waves with ground-based phase velocity equal to zero, $\lambda_Z$ is given as

$$\lambda_Z \approx \frac{2\pi \bar{U}_{||}}{N} \tag{16}$$

where $\bar{U}_{||}$ is the component of the mean wind speed parallel to the wave's horizontal wavenumber vector (Eckermann and Preusse, 1999). $\bar{U}_{||}$ is found by projecting the mean wind vector $\bar{U}$ in the direction of the wave propagation angle $\theta$ shown in Figs. 5(e) and 6(e).




Figures 5(f) and 6(f) show $\bar{U}_\parallel$ coincident with each granule at an altitude of 40 km from ECMWF operational analyses, projected onto each granule's regular grid. Orange contours and black arrows show the magnitude and direction of $\bar{U}_\parallel$ respectively. Vertical wavelength $\lambda_Z$ is shown in Figs. 5(g) and 6(g).

Towards the south-eastern corner of both granules, mean wind speeds become quite weak. As they fall below around 40 m s$^{-1}$, vertical wavelengths start to drop below the vertical resolution limit of the AIRS channel. The wave field may continue into this region, but the vertical wavelengths may be too short to be resolved such that wave amplitudes are attenuated to below the ambient noise level. This is particularly clear in the Antarctic Peninsula granule, where detectable wave fronts abruptly terminate just as $\bar{U}_\parallel$ begins to fall below 30 m s$^{-1}$.

In the South Georgia granule, peak momentum flux values of more than 500 mPa are associated with a small region of large amplitude and short horizontal wavelength wave features, located toward the south-eastern tip of the island. In the Antarctic Peninsula granule, momentum fluxes of a few hundred millipascals are generally co-located with the clearly visible wave structures in the raw brightness temperature perturbations just downwind of the peninsula. The magnitude, direction and distribution of momentum fluxes in both granules take reasonable values, and are broadly in line with previous AIRS gravity wave studies in the region (e.g Alexander and Teitelbaum, 2007; Alexander et al., 2009; Alexander and Teitelbaum, 2011).

The key strength of the results presented here is the much-improved spatial-spectral localisation and resolution capabilities provided by full two-dimensional treatment of the AIRS data. Confidence in the accuracy of subsequent measured quantities in our 2DST-based is thus greatly improved analysis over previous 1DST-based methods. Understandably, the former is more computationally intensive than the latter, and this should be considered if datasets are large or computational resources are limited.

## 5.4 Small-scale perturbations

In the implementation of any spectral image processing, it is important to strike a balance between accurate measurement of the desired properties and the spurious interpretation of noise. One of the advantages of AIRS measurements is the high horizontal resolution of the data. With the close exception of the Infrared Atmospheric Sounding Interferometer (IASI, e.g. Clerbaux et al., 2009), currently no other spaceborne instrument can measure stratospheric gravity waves with comparable horizontal resolution to AIRS. Therefore, accurate measurement of resolved waves with horizontal wavelengths close to the AIRS resolution limit are of great importance. Such short horizontal wavelength waves, if reliably resolved, will generally carry higher momentum fluxes via Eq. 15. However, as we approach the resolution limits of AIRS measurements and our spectral methods, our confidence in the accuracy of our measurement of such waves decreases.

In Fig. 5, our 2DST analysis resolves a very small region of short horizontal wavelengths over the south-eastern tip of South Georgia. Wave perturbations in this small region are just a few pixels across, but their uncorrected brightness temperature perturbations are large, with peaks of order 5 to 6 K. If these perturbations were located almost anywhere else on the granule we would likely attribute them to retrieval noise.

However, gravity waves with very short horizontal wavelengths, tightly packed in a region immediately downwind of a mountainous island, are in good agreement with mountain wave theory. Examples of such waves can be found in, for example,





the modelling studies of Shutts and Vosper (2011) and Alexander and Teitelbaum (2011). If these waves are indeed real, their accurate measurement is of great importance.

An added complication is introduced as a result of the mountain wave assumption used. Since these waves are only a few pixels across, their directions of propagation are difficult to define, introducing a random element. The component of
the mean wind parallel to the horizontal wavenumber vector can thus be very low, which decreases the vertical wavelength estimate, which in turn increases the attenuation correction applied to the observed temperature perturbations. This attenuation correction can increase temperature perturbations by 400% or more which, since momentum flux is proportional to the square of wave amplitude, can increase our estimate of momentum flux to extremely large values.

As a result, these very small-scale low-confidence perturbations can yield extremely high momentum fluxes which can
dominate the momentum budget of the entire granule if a mountain wave assumption is used, correctly or otherwise. This is evident in the South Georgia granule studied by Alexander et al. (2009) and shown in our Fig. 5, where peak momentum fluxes are localised over only a few large-amplitude pixels just over the south-eastern tip of the island. In Fig. 5(h), the flux peaks at almost 1000 mPa for one pixel in this small region, though the colour scale is saturated. Whether these fluxes are real or not, they can nevertheless be extremely large and should be approached with caution.

If these extremely high fluxes correspond to real waves, then their measurement is of critical importance. However, if such perturbations are simply instrument noise and their fluxes are spurious, then the biases and errors introduced by their inclusion in broader studies could be very large.

Without *a priori* knowledge of the wave environment, which is most readily gained by visual inspection of the AIRS measurements, it would be unwise to include the fluxes from these small-scale perturbations in any automated analysis. Indeed,
Wright et al. (2015b) suggest that when AIRS granules are pre-smoothed with a boxcar of width 3 pixels, resolved momentum fluxes calculated using the method of Alexander et al. (2009) can be reduced by a order of magnitude. This suggests that these small-scale, high-momentum flux features are reasonably common and can impact larger-scale momentum flux estimates, either realistically or spuriously.

Further work investigating this problem is encouraged. Pre-smoothing granules so as to exclude these perturbations (e.g.
Wright et al., 2015b) is one solution. Excluding even more wavenumber voices corresponding to short horizontal wavelengths from our 2DST analysis is another solution. However, in cases such as the South Georgia granule presented in Fig. 5, where such perturbations may well be physical, this exclusion can reduce peak momentum fluxes by orders of magnitude, introducing a systematic low-bias and thus further uncertainty.

## 6 Summary and Conclusions

In this study, we have applied the two-dimensional Stockwell transform (2DST) to granules of AIRS measurements, extracting gravity wave amplitudes, wavelengths and directions of propagation. Our 2DST method builds upon the work of Alexander and Barnet (2007), who used the one-dimensional Stockwell transform for the same purpose. Their method can introduce a strong cross-track bias problem, which we solve by using a full two-dimensional S-Transform.





We first define our 2DST implementation and test it on a specified wave field containing synthetic waves with known amplitudes, wavelengths and directions of propagation. We find that the 2DST provides very good spatial representation of the dominant spectral components of the specified wave field, accurately measuring wavelengths and orientations of all the synthetic waves.

Due to the spread of spectral power in the spectral domain and wave undersampling in the spatial domain, we find that localised wave amplitudes as measured by the 2DST are reduced by more than a factor of two when the typical Gaussian windowing function is used in the Stockwell transform. We compensate for this by decreasing the scaling parameter $c$ and by replacing the Gaussian window with alternative windowing functions, which we test on a granule of AIRS measurements over the southern Andes. We find that the use of an elliptical windowing function provides the best trade-off between spatial-spectral

localisation and the accurate measurement of wave amplitudes. Wave amplitude recovery is thus improved to around 80% to 90% of input values.

Next, we measure gravity wave amplitudes, horizontal wavelengths and directions of propagation in two granules of AIRS measurements over South Georgia and the Drake Passage/Antarctic Peninsula region. Our 2DST method significantly improves the two-dimensional representation of the dominant spectral features of the granules over previous 1DST methods. These spec-

tral features are directly measured in both dimensions simultaneously for each location of the granule, without the introduction of potential biases caused by the use of averaged co-spectra. This is a clear advantage over previous methods.

Another key advantage of our 2DST method is the ability to visually inspect the quality of our spectral analysis. By taking the real parts of the dominant localised spectral coefficients at each location, a "reconstruction" of the granule can be created. This can be used to fine-tune the adjustable parameters, and provide a useful sanity check on the performance of the 2DST.

Future work may involve comparing this output to the original data via a variance argument or similar, such that we can obtain a quantitative measure of the quality of the 2DST analysis for quality control purposes.

To conclude, our new 2DST-based gravity wave analysis method for AIRS data makes significant improvements over current methods in several key areas, and we would advocate its use in future work.

*Acknowledgements.* NPH is funded by a NERC studentship awarded to the University of Bath. CJW and NJM are supported by NERC grant

NE/K015117/1. The authors would like to thank the AIRS programme team for many years of hard work producing the data used.





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



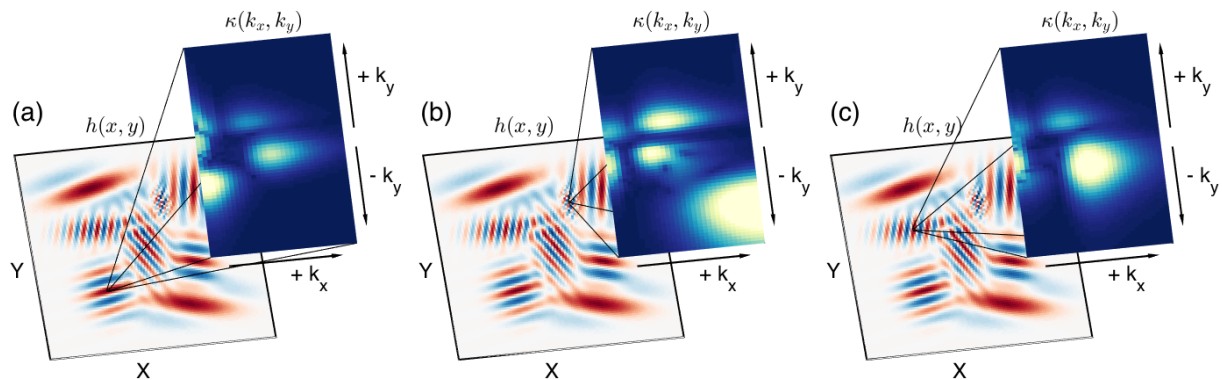

**Figure 1.** The specified wave field $h(x, y)$ (background) for which the two-dimensional Stockwell transform (2DST) has been computed. The absolute magnitudes of the localised 2DST wavenumber spectra $\kappa(k_x, k_y)$ (foreground) are plotted for three separate locations in panels (a), (b) and (c).

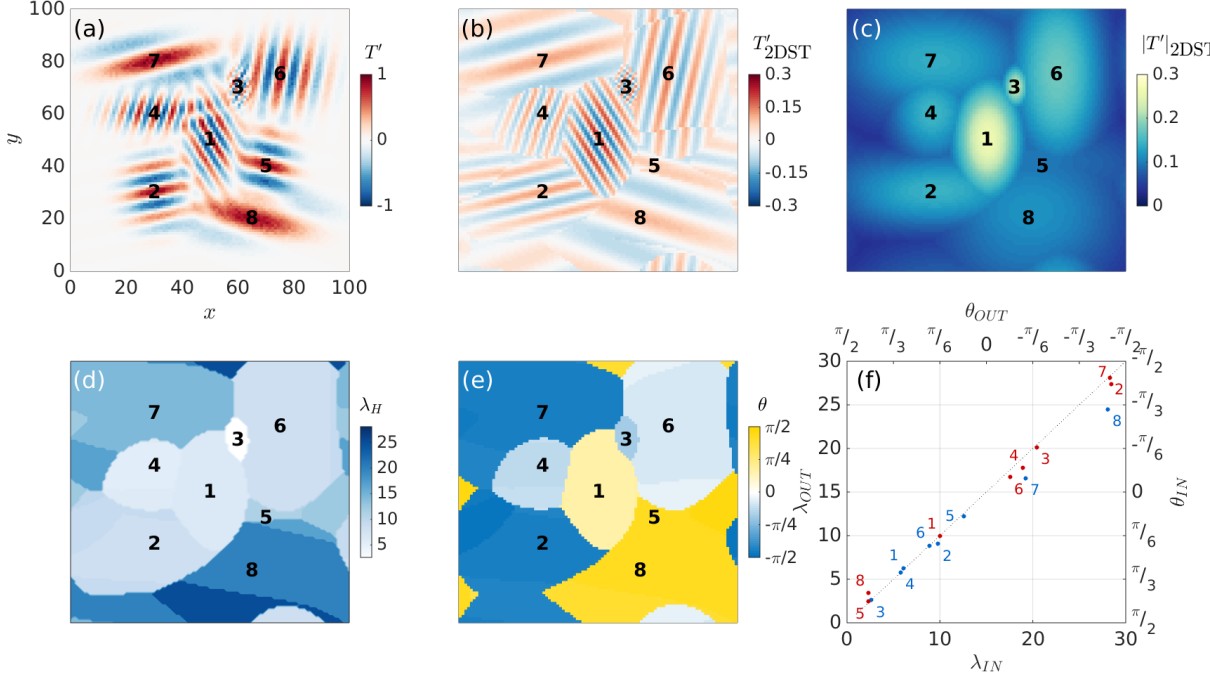

**Figure 2.** The specified wave field (a), containing synthetic waves numbered 1 to 8, for which our two-dimensional Stockwell transform analysis has been performed. The "reconstructed" wave field $T'_{2DST}$, instantaneous wave amplitudes $|T'|_{2DST}$, horizontal wavelengths $\lambda_H$ and directions of propagation $\theta$ (measured anticlockwise from the positive $x$ direction) are shown in panels (b), (c), (d), and (e) respectively. Distances and wavelengths have units of kilometres and amplitudes have units of Kelvin. Panel (f) compares input and measured wavelengths (blue numbered dots) and input and measured propagation angles (red numbered dots) for the eight synthetic waves. The dashed grey line in (f) shows 1:1 correspondence.




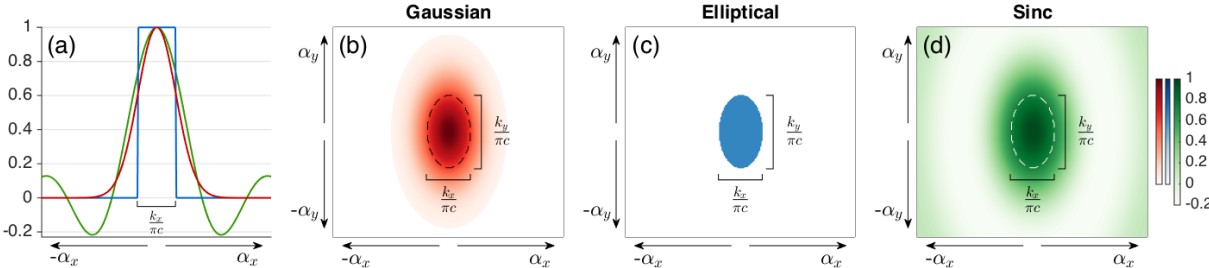

**Figure 3.** Panel (a) shows the central regions of the 1-D forms of the Gaussian (red), elliptical (blue), and sinc (green) windowing functions used for wavenumber localisation in the 2DST in Sect. 4. The 2-D forms of the Gaussian, elliptical, and sinc windows are shown in (b), (c) and (d) respectively, where $\alpha_x$ and $\alpha_y$ are translations in the spectral domain. Dashed ellipses in (b), (c) and (d) show the location of the $1\sigma$ contour of the 2-D Gaussian windowing function in Eq. 9 for arbitrary wavenumbers $k_x$ and $k_y$.

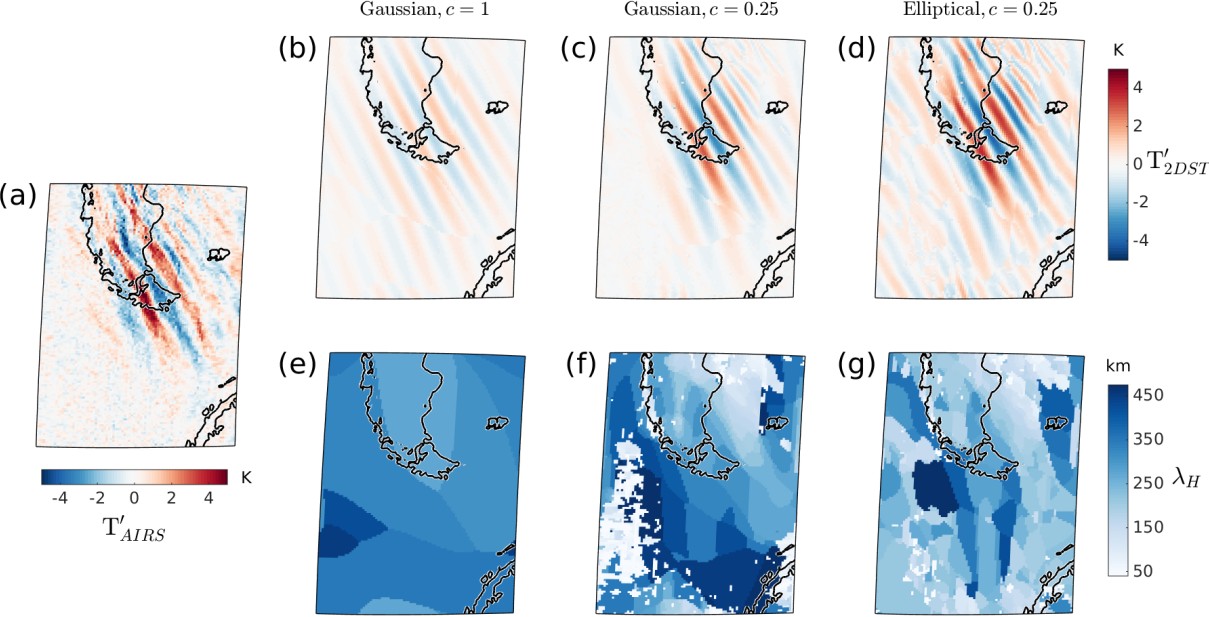

**Figure 4.** Orthographic projection of AIRS brightness temperature perturbations $T'_{AIRS}$ (a) from a granule over the Southern Andes at 0530 UTC on 24$^{\text{th}}$ May 2008, with "reconstructed" temperature perturbations $T'_{2DST}$ and horizontal wavelengths $\lambda_H$ computed using the 2DST with three different windowing approaches: (b,e) a Gaussian window with scaling parameter $c = 1$, (c,f) a Gaussian window with $c = 0.25$, and (d,g) an elliptical window with $c = 0.25$.



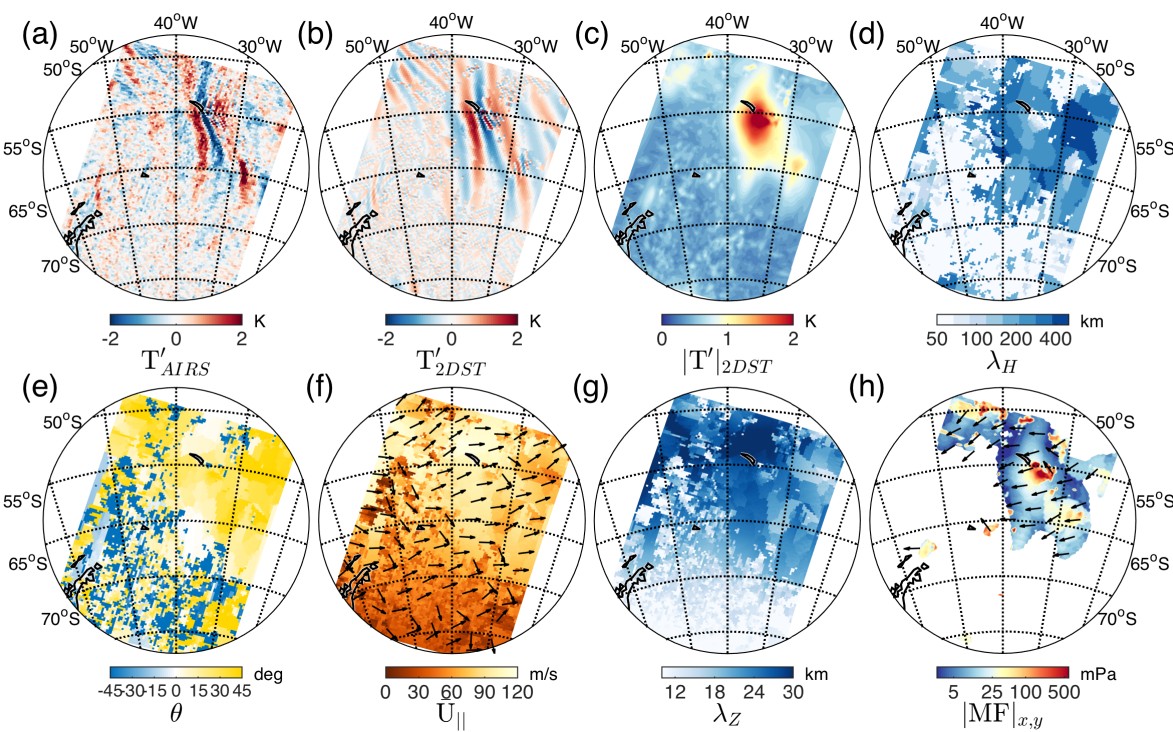

**Figure 5.** Orthographic projections of a granule of AIRS brightness temperature measurements $T'_{AIRS}$ (a) over South Georgia at around 0300 UTC on 6$^{th}$ September 2003 and selected outputs of our 2DST analysis (b-e). This granule was also analysed by Alexander et al. (2009) (their Fig. 3) using a one-dimensional S-Transform method. The 2DST outputs shown here are reconstructed brightness temperature perturbations $T'_{2DST}$ (b), instantaneous wave amplitude $|T'|_{2DST}$ (c), horizontal wavelength $\lambda_H$ (d) and wave propagation direction $\theta$ (e) in degrees anticlockwise from east. Also shown are mean wind speed parallel to the horizontal wavenumber vectors $\bar{U}_{||}$ (f) from ECMWF operational analyses at $z \approx 40$ km, vertical wavelength $\lambda_Z$ (g) and the magnitude of the horizontal component of vertical momentum flux $|MF|_{x,y}$ (h). Black arrows in (f) and (h) show the horizontal direction of $\bar{U}_{||}$ and $|MF|_{x,y}$ respectively. For details, see text.




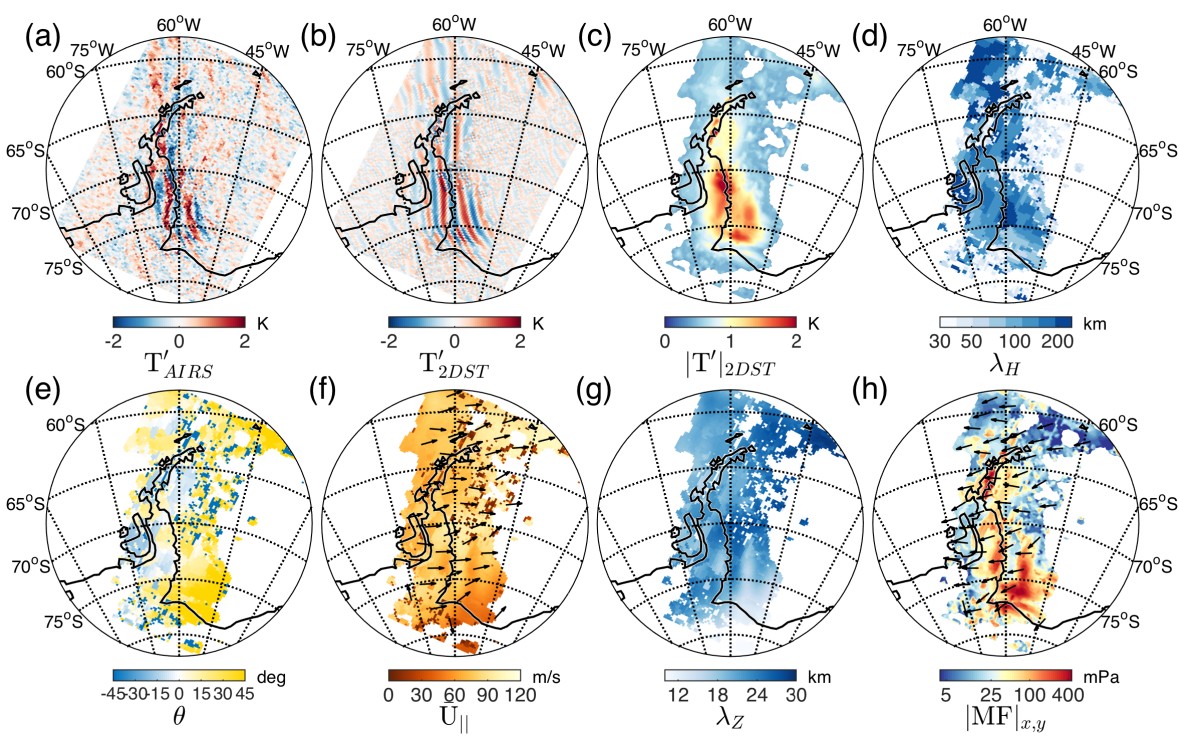

**Figure 6.** As Fig. 5, but for AIRS measurements over the Antarctic Peninsula around 0400 UTC on 2$^{nd}$ August 2010. These measurements were also analysed by Hoffmann et al. (2014, their Fig. 8) using a one-dimensional S-Transform method. Here, regions in panels (c–h) where the instantaneous wave amplitude $|T'|_{2DST}$ is less than one standard deviation below the mean are coloured white.