# Peer review of "A two-dimensional Stockwell Transform for gravity wave analysis of AIRS measurements"

_Atmospheric Measurement Techniques, 2015_

## Referee Comment (RC1) · Anonymous Referee #1 · 15 Feb 2016

**General Comment**

In their paper the authors introduce the 2D S-transform as a new method for deriving gravity wave amplitudes, horizontal wavelengths and propagation directions from brightness temperature distributions of the nadir-viewing Atmospheric Infrared Sounder (AIRS) instrument. The method is tested using simulated data, and optimized by selecting an elliptical window instead of the traditionally used Gaussian window. Advantages and disadvantages of the method are thoroughly discussed, and the method is applied to three granules of AIRS data over the Southern Andes, the Drake Passage/Antarctic Peninsula, and the isolated mountainous island of South Georgia.

Overall, this is a very interesting study and an important step forward in the estima-

tion of gravity wave properties and momentum fluxes. These parameters are needed to improve the representation of gravity waves in global models, which is one of the major uncertainties in global modeling. Only minor revisions are required before the manuscript is recommended for publication in AMT.

Main comments are:

(A) momentum fluxes without attenuation correction should also be shown

(B) a rough comparison with momentum fluxes from limb sounders should be included

Please find the Detailed Comments below.

**Detailed Comments**

(1) p.2, l.31:
for completeness, the reference Jiang et al., 2002 should be included:
Jiang, J. H., D. L. Wu, and S. D. Eckermann, Upper Atmosphere Research Satellite (UARS) MLS observation of mountain waves over the Andes, J. Geophys. Res., 107(D20), 8273, doi:10.1029/2002JD002091, 2002.

(2) p.3:
Sect. 1.1 is quite short and somehow out of place, in the introduction following the organization and overview of the paper. I would suggest to introduce a new section 2: "Data and method", shift Sect. 1.1 to 2.1, and change Sect. 2 to Sect. 2.2.

(3) Suggestion: briefly introduce the expression "voice" by stating that this denotes a specific wavenumber or frequency out of a discrete set.
(4) p.8, l.12: Please mention, if this is correct:
Similar to real-world waves, spread of frequencies would be expected also for your simulated waves because they form some kind of "wave packets" with the wave amplitude damped with increasing distance from the center of the packet. This will introduce a spread of spectral power that is not fully captured by just focusing on the dominant waves.

(5) p.8, l.23, and elsewhere:
It seems that in your paper wavenumbers are generally defined as 1/wavelength, rather than $2\pi$/wavelength.
This should be clarified and stated in the manuscript;
please check equations for consistency whether this definition of wavenumbers has effect on the different scaling factors that are required.

(6) Question about Fig.2:
If integrated over the whole domain, could it happen that the temperature variance contained in Fig.2b or Fig.2c could be higher than in Fig.2a? This information should also be included in the manuscript.

(7) Fig.3:
You should mention that the windowing functions are normalized with max. value of 1. This is different from the use in Eq.(7) where the integral over the windowing function should be unity.

(8) General comment:
Sometimes the windowing functions are given with scaling factors that make sure that the integral over this function should be unity. (Eq.(2)–(4)).
Sometimes it rather looks like the maximum of this function is scaled to unity (Eq.(5), (6), (8)–(11)). Is this correct?
Please comment/check for consistency.

(9) p.10/11, discussion of Fig.4:
As a cross check, please calculate the temperature variance over the whole AIRS
granule for the raw data (Fig.4a) and compare with the variances for the recon-
structed T' in Figs.4b, 4c, and 4d. Two reasons:
(1) It almost looks like the variance of the distribution in Fig.4(d) would overesti-
mate the raw data in Fig.4(a).
(2) Calculating these variances provides a rough measure which method per-
forms best in reproducing the total variance of the wave field.

(10) p.13, l.3-5:
Please mention that the resolution of this regularly-spaced grid closely matches
the spatial resolution of AIRS in the center of an AIRS swath, and not much in-
formation is lost. In the off-center regions of an AIRS swath the regularly-spaced
grid is even better than the spatial resolution of AIRS, however the grid will not
exactly match the location of the AIRS footprints.

(11) Fig.5/Fig.6:
Also momentum flux values without attenuation correction should be shown be-
cause this correction is quite substantial, possibly up to a factor of 100, and prob-
ably this correction is strongly susceptible to errors in the vertical wavelength.
This is particularly important because vertical wavelengths are derived assuming
that a mountain wave is observed, and based on ECMWF winds. At high altitudes
ECMWF winds can easily be biased by about 10m/s. There is some error dis-
cussion later in the manuscript, however these limitations should be more clearly
mentioned already in the discussion of Figs.5 and 6.
It should also be mentioned whether or not for Fig.5/Fig.6 an upper limit was used
for this correction.

(12) p.15, after l.9:
Overall, the agreement between Fig.5a and Fig.5b is very good.

However, to me it looks like the amplitude of the mentioned small-scale event in Fig.5b would be somewhat overestimated compared to Fig.5a. Further, in the reconstructed wave field the strongest positive wave crest of this small-scale event is somewhat shifted to the west, into the negative wave phase of the larger scale event. This is not seen in the original AIRS brightness temperatures.

Another difference is seen in Figs.5a and 5b at 63S, 28W where in the original brightness temperatures only a larger scale bow shaped positive anomaly is seen. In the reconstructed T'-field for the dominant spectral features, however, there is a smaller amplitude shorter scale feature.

Therefore you should mention that, apart from the really good agreement between Fig.5a and Fig.5b, there are still some remaining uncertainties.

(13) p.15, after l.14:
At least a rough comparison with gravity wave momentum fluxes from limb sounders should be included. To me it looks like typical values for limb sounders over the Antarctic Peninsula during July are somewhere in the range 1–10mPa (Geller et al., 2013; Ern et al., 2011, their Figs.3c and 9d). These values do not include attenuation corrections, still your AIRS values seem to be much higher, which is an important information.

Reference:
Ern, M., P. Preusse, J. C. Gille, C. L. Hepplewhite, M. G. Mlynczak, J. M. Russell III, and M. Riese (2011), Implications for atmospheric dynamics derived from global observations of gravity wave momentum flux in stratosphere and mesosphere, J. Geophys. Res., 116, D19107, doi:10.1029/2011JD015821.

**Technical Comments**

- p.1, l.7:
  and directions in both directions simultaneously. → in both directions simultaneously, and thereby the propagation direction of the waves.

- p.3, l.20: fluxes easier → fluxes are easier

- p.3, l.20: pertubrations → perturbations

- p.6, l.22: descirbes → describes

- Fig.2c:
  In Fig.2c spectral amplitudes are shown. In the manuscript, this parameter is denoted $\xi$, while the parameter given at the colorbar in Fig.2c is denoted $|T'|$. This is somehow misleading because temperature fluctuations were denoted T' before.

- p.8, l.23:
  probably, this should read $(k_x^2 + k_y^2)^{-0.5}$
  (lowercase k, and power of $-0.5$, instead of $-2$)

- p.12, l.6: furture → future

---

## Referee Comment (RC2) · Anonymous Referee #2 · 28 Feb 2016

**A review report on "A two-dimensional Stockwell Transform for gravity wave analysis of AIRS measurements" by N. P. Hindley, N. D. Smith C. J. Wright, and N. J. Mithell**

Recommendation: Minor revision

**General comments:**

The paper presents usability of a 2D Stockwell transform (S-transform) for gravity wave analysis using nadir-view satellite measurements. Most previous studies use onedimensional S-transform for cross-track rows and cross-spectra between adjacent crosstrack rows to estimate two dimensional horizontal wavenumbers for observed waves. However, this method has a problem that the accuracy of along-track horizontal wavenumber estimates is generally low compared with that of cross-track ones. The 2D S-transform is a natural method to overcome this problem. The 2D S-transform is not now in a sense that it has been applied to other fields than atmospheric science. However, this paper carefully examine the usability of the 2D S-transform for a particular application of gravity wave parameter estimation from AIRS observations, such as limitation of the method or implication of the results obtained from the method for very long or very short horizontal wavelength waves. They even propose variations of window functions other than a standard Gaussian shape to estimate more accurate (or reasonable) amplitudes. This argument is quite interesting. The manuscript is well written and the points for discussion are clear. Thus I think that this paper has a value to be published in Atmos. Meas. Tech. However, the current manuscript contains several issues that need minor but important revision before being accepted for publication.

**Comments**

p.3 1.20: Momentum flux estimation from temperature fluctuations is not easy. Most methods such as shown in (15) use an assumption that the wave fields are monochromatic. Vertical wavelengths are hardly estimated from nadir view observations as discussed in this manuscript, too. There is also an observational window problem: AIRS cannot observe waves with short vertical wavelengths and/or very long horizontal wavelengths.

p.4, l.23: A reference or a short description of derivation for (5) should be useful (e.g. convolution theorem).

p.5 l.1:  $H(t) \rightarrow h(t)$

p.14, l.1: For AIRS observation, the angle between along-track direction and across-trck direction is not necessarily 90°. Thus the formula for absolute horizontal wavelengths is not correct. This formula should be written using the angle between the directions.

p.14, l.15: Please specify underlying assumptions for (15). This formula is valide for monochromatic and hydrostatic gravity waves which are not influenced by the Coriolis effect.

---

## Author Comment (AC3) · 5 May 2016

**Correction to authors' response to Anonymous Reviewer #1 : A two-dimensional Stockwell Transform for gravity wave analysis of AIRS measurements *doi:10.5914/amt-2015-383**

Neil P Hindley[1]

[1]Centre for Space, Atmosphere and Ocean Science, University of Bath, UK

*Correspondence to:* Neil Hindley (n.hindley@bath.ac.uk)

We would like to submit a correction to our response to Anonymous Reviewer #1 for the manuscript *amt-2015-383*.

**1.** The axes of Fig. R1 are labelled incorrectly. The correctly labelled figure is shown below:

[Figure]

**Figure R1.** Scatter plot (blue dots) of total brightness temperature perturbation variance of AIRS granules $\sigma_A^2$ globally at $\sim$41 km from June-August 2010 against temperature variance of "reconstructed" temperature perturbations using a two-dimensional Stockwell tranform analysis method. The black dashed line shows where $\sigma_A^2 = \sigma_R^2$.

---

## Author Comment (AC2)

**Authors' response to Anonymous Reviewer #2 : A two-dimensional Stockwell Transform for gravity wave analysis of AIRS measurements *doi:10.5914/amt-2015-383**

Neil P Hindley[1]

[1]Centre for Space, Atmosphere and Ocean Science, University of Bath, UK

*Correspondence to:* Neil Hindley (n.hindley@bath.ac.uk)

We would like to thank the anonymous reviewer for their very helpful suggestions. These will greatly contribute to the improvement of the manuscript, and we have applied all changes as requested.

A LatexDiff document, denoting all changes to the manuscript, is attached at the end of these responses.

**5 A note to reviewers: Correction in original submission**

In the first submission of this manuscript, we introduced a new spectral windowing function for the two-dimensional Stockwell transform: the elliptical window. In Sect. 4.4 of this first submission (p.11, l.20) we stated that this new window is only partially valid for use in the Stockwell transform.

We have since been able to show that this new window is in fact fully valid for use in the Stockwell transform. We did this by

10 showing that the integral of its spatial form is equal to unity. A short proof of this is provided in an Appendix in the revised manuscript. This greatly strengthens the integrity of the study and will be of use to others in the future. The results of the study are unchanged.

Accordingly, we would like to add D. Andrew S. Rees, at the University of Bath, to the list of authors of this manuscript for his contribution to this correction.

15 In the revised manuscript, we refer to the elliptical window as the Elliptic-Bessel window, which more appropriately describes its wavenumber and spatial forms. The Sinc window, which was mentioned only in passing and not used in the study, is now redundant and has been removed from the revised manuscript. We have updated Figure 3 to reflect this.

**Specific comments and authors' responses**

(1) **Reviewer's comment:** *p.3 l.20: Momentum flux estimation from temperature fluctuations is not easy. Most methods such as shown in (15) use an assumption that the wave fields are monochromatic. Vertical wavelengths are hardly estimated from nadir view observations as discussed in this manuscript, too. There is also an observational window problem: AIRS cannot observe waves with short vertical wavelengths and/or very long horizontal wavelengths.*

   – **Authors' response:** We agree that limitations remain, which are discussed in more detail in Section 5. We have added a short acknowledgement of this to the paragraph in question.

(2) **Reviewer's comment:** *p.4, l.23: A reference or a short description of derivation for (5) should be useful (e.g. convolution theorem).*

   – **Authors' response:** Reference and short description added. The full derivation of Eqn. 5, which does indeed involve the convolution theorem, can be found in (Stockwell, 1999, their Section 3.5.1), and applications of this spectral formulation can also be found in Mansinha et al. (1997a, b); Pinnegar and Mansinha (2003); Stockwell (2007) and others.

(3) **Reviewer's comment:** *Suggestion: briefly introduce the expression "voice" by stating that this denotes a specific wavenumber or frequency out of a discrete set.*

   – **Authors' response:** Added as suggested in Section 2. The "voice" expression is now used throughout the manuscript, where applicable.

(4) **Reviewer's comment:** *p.5 l.1: $H(t)$ -> $h(t)$*

   – **Authors' response:** Fixed.

(5) **Reviewer's comment:** *p.14, l.1: For AIRS observation, the angle between along-track direction and across-track direction is not necessarily $90°$. Thus the formula for absolute horizontal wavelengths is not correct. This formula should be written using the angle between the directions.*

   – **Authors' response:** We do not quite agree:

   In the frame of reference of the AIRS instrument aboard the Aqua satellite, viewing at the nadir the angle between the along-track direction and each cross-track scan is always $90°$, or as close as physically possible. It is only when the measurement swath is projected onto the ellipsoidal surface of the earth that this angle can vary towards the edges of the swath and may not necessarily be $90°$ as the reviewer suggests.

However, the geolocation information contained within the AIRS measurements allows us, to a good approximation, to interpolate this irregularly-sampled projection back into a regular and orthogonal grid, as viewed by the AIRS instrument. In this grid, the along-track and cross-track directions are thus orthogonal, and the equation on p.14 is valid for measuring projection of these waves in the AIRS granule.

We do acknowledge however that some other uncertainties may arise. Near to the poles for example, small "warping" effects may occur where the rotation of the earth under the Aqua satellite during a scan is non-negligible, such that the image is not instantaneous for the geographic region it views.

(6) **Reviewer's comment:** *p.14, l.15: Please specify underlying assumptions for (15). This formula is valide for monochromatic and hydrostatic gravity waves which are not influenced by the Coriolis effect.*

– **Authors' response:** As requested, we have now specified the assumptions for Eqn. 15 in Section 5.3. We have added the text: "Eqn. 15 assumes the mid-frequency approximation (Fritts and Alexander, 2003, their Sect. 2.1), which is valid for large portion of the observable gravity wave spectrum. For these waves, $m^2 \ll (k^2 + l^2)$ and Coriolis effects are neglible, as is the case for nearly all waves measured here due to the size of the AIRS beam footprint (Hoffmann et al., 2014) and the vertical weighting function of the $667.77\,\mathrm{cm}^{-1}$ channel (Alexander and Barnet, 2007). When compared with momentum fluxes calculated without making the mid-frequency approximation, Ern et al. (2004) found discrepancies typically not exceeding around 10%. "

**References**

Alexander, M. J. and Barnet, C.: Using satellite observations to constrain parameterizations of gravity wave effects for global models., J. Atmos. Sci., 64, 1652–1665, doi:10.1175/JAS3897.1, 2007.

Ern, M., Preusse, P., Alexander, M. J., and Warner, C. D.: Absolute values of gravity wave momentum flux derived from satellite data, J. Geophys. Res., 109, D20 103, doi:10.1029/2004JD004752, 2004.

Fritts, D. C. and Alexander, M. J.: Gravity wave dynamics and effects in the middle atmosphere, Reviews of Geophysics, 41, 1003, doi:10.1029/2001RG000106, 2003.

Hoffmann, L., Alexander, M. J., Clerbaux, C., Grimsdell, A. W., Meyer, C. I., Roessler, T., and Tournier, B.: Intercomparison of stratospheric gravity wave observations with AIRS and IASI, Atmos. Meas. Tech., 7, 4517–4537, doi:10.5194/amt-7-4517-2014, 2014.

Mansinha, L., Stockwell, R. G., and Lowe, R. P.: Pattern analysis with two-dimensional spectral localisation: Applications of two-dimensional S transforms, Physica A., 239, 286–295, doi:10.1016/S0378-4371(96)00487-6, Proceedings of the International Conference on Pattern Formation in Fluids and Materials CPiP 96 (Collective Phenomena in Physics 96), University of Western Ontario, London, Canada, June 13-15, 1996, 1997a.

Mansinha, L., Stockwell, R. G., Lowe, R. P., Eramian, M., and Schincariol, R. A.: Local S-spectrum analysis of 1-D and 2-D data, Physics of the Earth and Planetary Interiors, 103, 329–336, doi:10.1016/S0031-9201(97)00047-2, Conference on Geonomy in honor of Professor J A Jacobs, Edinburgh, Scotland, June 03-04, 1996, 1997b.

Pinnegar, C. R. and Mansinha, L.: The S-transform with windows of arbitrary and varying shape, Geophysics, 68, 381–385, doi:10.1190/1.1543223, 2003.

Stockwell, R. G.: "S-Transform Analysis of Gravity Wave Activity from a Small Scale Network of Airglow Imagers", Ph.D. thesis, "University of Western Ontario", 1999.

Stockwell, R. G.: A basis for efficient representation of the S-transform, Digital Signal Processing, 17, 371 – 393, doi:http://dx.doi.org/10.1016/j.dsp.2006.04.006, 2007.

**A two-dimensional Stockwell Transform for gravity wave analysis of AIRS measurements**

Neil P. Hindley[1], Nathan D. Smith[1], Corwin J. Wright[1], D. Andrew S. Rees[2], and Nicholas J. Mitchell[1]

[1]Centre for Space, Atmosphere and Ocean Science, University of Bath, UK
[2]Department of Mechanical Engineering, University of Bath, UK

*Correspondence to:* Neil Hindley (n.hindley@bath.ac.uk)

**Abstract.** Gravity waves (GWs) play a crucial role in the dynamics of the earth's atmosphere. These waves couple lower, middle and upper atmospheric layers by transporting and depositing energy and momentum from their sources to great heights. The accurate parametrization of GW momentum flux is of key importance to general circulation models but requires accurate measurement of GW properties, which has proved challenging. For more than a decade, the nadir-viewing Atmospheric Infrared Sounder (AIRS) aboard NASA's Aqua satellite has made global, two-dimensional (2-D) measurements of stratospheric radiances in which GWs can be detected. However, one problem with current one-dimensional methods for GW analysis of these data is that they can introduce significant unwanted biases. Here, we present a new analysis method that resolves this problem. Our method uses a 2-D Stockwell transform (2DST) to measure GW amplitudes, horizontal wavelengths and directions of propagation using both the along-track and cross-track dimensions simultaneously. We first test our new method and demonstrate that it can accurately measure GW properties in a specified wave field. We then show that by using a new elliptical spectral window in the 2DST, in place of the traditional Gaussian, we can dramatically improve the recovery of wave amplitude over the standard approach. We then use our improved method to measure GW properties and momentum fluxes in AIRS measurements over two regions known to be intense hot spots of GW activity: (i) the Drake Passage/Antarctic Peninsula and (ii) the isolated mountainous island of South Georgia. The significance of our new 2DST method is that it provides more accurate, unbiased and better localised measurements of key GW properties compared to most current methods. The added flexibility offered by the scaling parameter and our new spectral window presented here extend the usefulness of our 2DST method to other areas of geophysical data analysis and beyond.

**1 Introduction**

Gravity waves are a vital component of the atmospheric system. These propagating mesoscale disturbances can transport energy and momentum from their source regions to great heights. They thus are a key driving mechanism in the dynamics of 
[revised manuscript text omitted]

The weighting function of the 667.77 cm$^{-1}$ channel peaks near 3 hPa ($\sim$40 km), with a full width at half maximum of $\sim$12 km (Alexander and Barnet, 2007, also illustrated in Figure 1 of W Gravity waves with vertical wavelengths shorter than 12 km are thus unlikely to be resolved and vertical wavelengths close to this limit will be strongly attenuated.

If the vertical wavelength is known, it is possible to correct for this attenuation by dividing the amplitude by an appropriate rescaling factor (Alexander and Barnet, 2007, their Fig. 4). Although methods for measuring long vertical wavelengths using multiple AIRS channels have been developed (e.g. Hoffmann and Alexander, 2009), we do not have direct measurements of vertical wavelength from our single AIRS channel, and so we do not apply such a correction to brightness temperature perturbations at this stage. The true amplitude of some waves in our initial analysis may therefore be between two and five times greater than the values shown.  Later however, for the estimation of momentum flux in Sect. 5.3, we do apply the attenuation correction described in Alexander and Barnet (2007).

**3**

**2.1 The Stockwell transform**

In its analytical form, the one-dimensional Stockwell transform (Stockwell et al., 1996) closely resembles a continuous wavelet transform (CWT) with a complex sinusoidal mother wavelet windowed with a scalable Gaussian window (Gibson et al., 2006). For time series data, this scalable Gaussian localises wave perturbations in the time domain through spectral localisation in the frequency domain.

For a smoothly-varying, continuous and one-dimensional function of time $h(t)$, the generalised analytical form of the S-transform $S(\tau, f)$ (e.g. Pinnegar and Mansinha, 2003) is given as

$$S(\tau, f) = \int\limits_{-\infty}^{\infty} h(t)\, \omega_g(t - \tau - t, f)\, e^{-i2\pi f t}\, dt, \qquad (2)$$

where $\tau$ is translation in the time domain, $f$ is frequency and  $\omega_g(t-\tau,f)$ is a windowing function, scaled with frequency, that provides spatial and spectral localisation. Traditionally,  $\omega_g(t-\tau,f)$ takes the form of the normalised Gaussian window

$$\omega_{gaug}(\underbrace{t-\tau}_{} \underbrace{-t}_{},f) = \frac{|f|}{c\sqrt{2\pi}} \underbrace{\frac{1}{\sigma\sqrt{2\pi}}}_{} e^{\underbrace{\frac{-(t-\tau)^2 f^2}{2c^2}}_{} \underbrace{\frac{-(t-\tau)^2}{2\sigma^2}}_{}} \tag{3}$$

5     where $\sigma$ is the standard deviation. A key aspect of the Gaussian window  in Eqn. 3 is that the standard deviation is scaled for each frequency as $\sigma = \frac{c}{|f|}$

$$S(\tau,f) = \frac{|f|}{c\sqrt{2\pi}} \int\limits_{-\infty}^{\infty} h(t) e^{-\frac{(t-\tau)^2 f^2}{2c^2}} e^{-i2\pi ft} dt,$$

 , where $c$ is a scaling parameter
10    usually set to 1 (Mansinha et al., 1997a). This window is often referred to as the "voice Gaussian", which provides localisation of a specific frequency "voice" (Stockwell, 1999). Another key aspect of the Gaussian in Eqn. 3 is the normalisation factor $1/\sigma\sqrt{2\pi}$, which ensures that the integral of the window over all $t$ is equal to unity, a requirement for any windowing function used in the S-transform. Substituting Eqn. 3 into Eqn. 2 allows us to write the S-transform more explicitly as

$$S(\tau,f) = \frac{|f|}{c\sqrt{2\pi}} \int\limits_{-\infty}^{\infty} h(t) e^{-\frac{(t-\tau)^2 f^2}{2c^2}} e^{-i2\pi ft} dt. \tag{4}$$

15   Typically, the scaling parameter $c$ is set to 1 (e.g. Stockwell et al., 1996; Alexander et al., 2008; Wright and Gille, 2013), but  it may also be set to other values to  achieve more specific time-frequency localisation requirements (e.g. Mansinha et al., 1997b; Fritts et al., 1998; Pinnegar and Mansinha, 2003). Setting $c > 1$ provides enhanced frequency localisation at the expense of time localisation, and contrarily setting $c < 1$ achieves enhanced time localisation at the expense of frequency localisation.  This effect is discussed in more detail in Sect. 4.

20   To compute the S-transform using the form in  Eqn. 4, it seems we must compute  a convolution involving the voice Gaussian and the time series for each frequency voice $f$.  , which can become quite computationally intensive. (Stockwell, 1999, their Sect. 3.5.1) showed that under the convolution theorem (Brigham, 1974), the time-domain convolution in Eqn. 4 could be written as a frequency-domain multiplication as

25   $$S(\tau,f) = \int\limits_{-\infty}^{\infty} H(\alpha+f) e^{-\frac{2\pi^2 c^2 (\alpha-f)^2}{f^2}} \underbrace{\frac{-2\pi^2 c^2 \alpha^2}{f^2}}_{} e^{i2\pi\alpha\tau} d\alpha \tag{5}$$

 where $H(\alpha + f)$ is a shifted version of $H(\alpha)$, which is in turn the frequency analogue of $H(t)$. The

$$\omega_{gau}(\alpha - f, f) = e^{\frac{-2\pi^2 c^2 (\alpha - f)^2}{f^2}}$$

 frequency-domain form of the voice Gaussian, denoted by $\omega_g(\alpha, f)$, is given as

$$\omega_g(\alpha, f) = e^{\frac{-2\pi^2 c^2 \alpha^2}{f^2}} \tag{6}$$

The standard deviation $\sigma_\alpha$ of this frequency-domain Gaussian window in Eqn. 6 scales with frequency as $\sigma = \frac{|f|}{2\pi c}$. $\sigma_\alpha = |f|/c$. Note that this voice Gaussian is unnormalised; its peak value is equal to 1 in the frequency domain.

In this frequency-domain form, the S-transform is computed for each frequency voice $f$ as the inverse Fourier transform of the product of $H(\alpha)$ $H(\alpha + f)$ and the corresponding frequency-domain voice Gaussian $\omega_g(\alpha, f)$ in Eqn. 6.  Crucially, writing the S-transform  as the frequency-domain multiplication  in Eqn. 5 enables computationally efficient ("fast") discrete Fourier transform (DFT) algorithms  and simple multiplication operations to be used. The S-transform is most commonly implemented in this manner within the atmospheric sciences.

The S-transform has a number of desirable characteristics for geophysical data analysis. Unlike a CWT, the absolute magnitudes of the complex-valued S-transform coefficients in $S(\tau, f)$  are directly related to the true underlying amplitude of the corresponding frequency voice $f$ at each location $\tau$. Information regarding wave amplitude is not strictly recoverable from a CWT, since the corresponding CWT coefficients are psuedo-correlation coefficients between the signal and the analysing wavelet.

One disadvantage to using fast DFT algorithms in an S-Transform implementation is the familiar coarse wavelength resolution at low frequencies, a limitation not encountered by the CWT. Since both the S-transform and DFT algorithms are easily extended to higher dimensions however, the reduced computational expense of a DFT-based S-transform makes this  a practical tool for large 2-D datasets. Retention of the wave amplitude information in the S-transform is another key advantage.

**2.2 The two-dimensional Stockwell transform**

The S-transform is easily extended to higher dimensions. For a two-dimensional 2-D image $h(x, y)$, the two-dimensional S-transform (2DST) is given by (Mansinha et al., 1997a; Stockwell, 1999)

$$S(\tau_x, \tau_y, kf_x, kf_y) = \frac{|k_x||k_y|}{2\pi c^2} \times \frac{|f_x||f_y|}{2\pi c^2} \int_{-\infty}^{\infty} \int_{-\infty}^{\infty} h(x, y) e^{-\frac{(x-\tau_x)^2 k_x^2 + (y-\tau_y)^2 k_y^2}{2c^2} - \left(\frac{(x-\tau_x)^2 f_x^2 + (y-\tau_y)^2 f_y^2}{2c^2}\right)} e^{-i2\pi(k_x x + k_y y)} dx dy^{-i2\pi(f_x x + f_y}$$

$$\tag{7}$$

where $\tau_x$, $\tau_y$  are translation in the $x$ and $y$ directions respectively. Here, $f_x$ and $k_y$ are translation and wavenumber $f_y$ are simple spatial frequencies (inverse of wavelength) in the $x$ and $y$ directions respectively.

, following the notation of Stockwell (1999). For the remainder of the present paper however, we switch to using angular wavenumbers $k_x = 2\pi f_x$ and $k_y = 2\pi f_y$, since this notation is more commonly used in the atmospheric sciences. Rewriting Eqn. 7 in terms of angular wavenumbers $k_x$ and $k_y$ gives

$$S(\tau_x, \tau_y, k_x, k_y) = \frac{|k_x||k_y|}{8\pi^3 c^2} \int\limits_{-\infty}^{\infty} \int\limits_{-\infty}^{\infty} h(x,y)\, e^{-\left(\frac{(x-\tau_x)^2 k_x^2 + (y-\tau_y)^2 k_y^2}{8\pi^2 c^2}\right)} e^{-i(k_x x + k_y y)}\, dx\, dy \tag{8}$$

The Gaussian windowing term in Eqn. 8 describes the 2-D voice Gaussian $w_g(x - \tau_x, y - \tau_y, k_x, k_y)$, where

$$w_g(x, y, k_x, k_y) = \frac{|k_x||k_y|}{8\pi^3 c^2}\, e^{-\frac{k_x^2 x^2 + k_y^2 y^2}{8\pi^2 c^2}} \tag{9}$$

This is the 2-D form of the 1-D Gaussian window in Eqn. 3. Here, the  standard deviations of the 2-D Gaussian window in Eqn. 9 are scaled with wavenumber in the $x$  $y$ directions as $2\pi c/|k_x|$ and $2\pi c/|k_y|$, where $c$ is  a scaling parameter.

As  discussed in Sect. 2.1, greater computational efficiency  is achieved by computing the 2DST as an operation in the wavenumber domain as

$$S(\tau_x, \tau_y, k_x, k_y) = \frac{1}{4\pi^2} \int\limits_{-\infty}^{\infty} \int\limits_{-\infty}^{\infty} H(\alpha_x + k_x, \alpha_y + k_y) \times \dots\, e^{-2\pi^2 c^2 \left(\frac{(\alpha_x - k_x)^2}{k_x^2} + \frac{(\alpha_y - k_y)^2}{k_y^2}\right) - \left(\frac{2\pi^2 c^2 \alpha_x^2}{k_x^2} + \frac{2\pi^2 c^2 \alpha_y^2}{k_y^2}\right)} e^{i2\pi(\alpha_x \tau_x + \alpha_y \tau_y) i(\alpha_x \tau_x + \alpha_y \tau_y)} d \tag{10}$$

where  $H(\alpha_x + k_x, \alpha_y + k_y)$ is a shifted version of $H(\alpha_x, \alpha_y)$, which is in turn the wavenumber analogue of the input image $h(x,y)$.

Here wavenumbers $k_x$ and $k_y$ are used to scale, in $\alpha_x$ and $\alpha_y$ directions respectively, the standard deviations of the wavenumber-domain form of the 2-D voice Gaussian $W_g(\alpha_x, \alpha_y, k_x, k_y)$, which is given as

$$W_g(\alpha_x, \alpha_y, k_x, k_y) = e^{-\left(\frac{2\pi^2 c^2 \alpha_x^2}{k_x^2} + \frac{2\pi^2 c^2 \alpha_y^2}{k_y^2}\right)} \tag{11}$$

The 2DST  is introduced and well-described by Mansinha et al. (1997a) and Mansinha et al. (1997b), who demonstrated its promise for pattern analysis. It has since been discussed and  applied in a variety of fields  to our knowledge it has yet to be  used for geophysical data analysis in the atmospheric sciences, despite the wide use of the 1-D form. In the following section, we describe our 2DST implementation methodology for the purpose of gravity wave analysis from 2-D data.

**3  2DST  analysis of a specified wave field**

To assess the capabilities of the 2DST, it is logical to first apply it to a two-dimensional specified wave field containing synthetic waves with known characteristics.

We create a specified wave field $h(x,y)$ with dimensions $100 \times 100$ km containing synthetic waves with unit  amplitudes and known wavelengths. Wave amplitudes are defined as temperature perturbations $T'$ in units of Kelvin. The synthetic waves are localised around their central locations with Gaussian functions (although note that they do overlap). We also add random ("salt and pepper") noise up to 10% of the wave amplitude.

We first compute the 2-D DFT $H(\alpha_x, \alpha_y)$ of our specified wave field $h(x,y)$. To recover an estimate of  the underlying wave amplitude, we use the familiar symmetry around the zeroth frequency in the Fourier domain to recover a 2-D analogy of the analytic signal, following the approach of Stockwell (1999). A 2-D DFT contains four quadrants that contain coefficients which are in complex-conjugate pairs with the coefficients in the opposite quadrant. The sum of these pairs always yields a real signal. By setting the coefficients of two of these quadrants to zero, and doubling their opposite quadrants, we obtain a complex-valued image when we take the inverse DFT. The magnitude of this image is  analogous to the underlying wave amplitude, while the complex part  describes instantaneous phase. All coefficients not in a complex conjugate pair are unchanged.

$$\omega_{gau}(\alpha_x, \alpha_y, k_x, k_y) = e^{-2\pi^2 c^2 \left( \frac{(\alpha_x - k_x)^2}{k_x^2} + \frac{(\alpha_y - k_y)^2}{k_y^2} \right)},$$

 The full 2DST spectrum $S(\tau_x, \tau_y, k_x, k_y)$ can then be computed by taking the inverse 2-D DFT of the product of the shifted spectrum $H(\alpha_x - k_x, \alpha_y - k_y)$ and the corresponding voice Gaussian $W_g(\alpha_x, \alpha_y, k_x, k_y)$ for each wavenumber voice $k_x$ and $k_y$ .

For our purposes, we do not need to evaluate $S(\tau_x, \tau_y, k_x, k_y)$ for all positive and negative wavenumbers, since evaluating all positive and negative values of $k_y$ and only the positive values of $k_x$ gives us all the degrees of freedom. There is a residual $180°$ ambiguity in wave propagation direction which cannot be broken without additional information, which is supplied in Sect. 5.3.

[revised manuscript text omitted]

We suspect the main reason for the reduced amplitudes relates to the "spreading" of spectral power in the transform.  Here, as is often the case for gravity waves in the real world, our simulated waves form small wave packets, where wave amplitude decreases around a central location. Such wave packets are usually represented in the spectral domain as some combination of wavenumber voices,  in addition to the dominant wavenumber of each of the

5 packets, in order to accurately describe their spatial properties. This means that the spectral power of a single, non-infinite wave packet can be spread across multiple wavenumber voices. Spectral leakage  can further contribute to this effect.

The Gaussian window in the 2DST is equal to one at its central location, but immediately falls away with increasing radius. This means that  any spectral power contained in adjacent wavenumber voices, which is required to

10 fully reconstruct the wave, is reduced. When the inverse DFT is computed, the recovered wave amplitude at this location is thus often diminished.

 A further reason for the diminished amplitude recovery in Fig. 2(b) is due to wave undersampling. This undersampling effect is worse for longer wavelengths, since fewer wave cycles are present in the same-sized region of the image. The wave undersampling limitations of the S-transform are well-understood in one dimension (Wright, 2010; Wright et al., 2015b).

15 Figure 2(c) shows the absolute magnitude of  the complex-valued image $|\xi(\tau_x, \tau_y)|$, which corresponds to the  full underlying amplitude of the dominant wave at each location. This output is useful for defining regions of the specified wave field that do or do not  contain clear and obvious wave features (McDonald, 2012).

The horizontal wavelength  $\lambda_H(\tau_x, \tau_y) = (K_x(\tau_x, \tau_y)^2 + K_y(\tau_x, \tau_y)^2)^{-1/2}$ of the dominant

20 wave at each location is shown in Fig. 2(d). Again, the different regimes of each wave in the specified wave field are clearly distinguished.

The direction of wave propagation $\theta(\tau_x, \tau_y)$, measured anticlockwise from the x-axis, is found as $\tan^{-1}\frac{K_x}{K_y}$, and plotted as Fig. 2(e). Note that $\theta(\tau_x, \tau_y)$ is subject to a $\pm\pi$ radian ambiguity, which is reconciled with *a priori* information in our AIRS analysis in Sect. 5.3.

25 To assess the effectiveness of our spectral analysis of the specified wave field, we compare the known wavelengths and propagation angles of the synthetic waves in the test image with the 2DST-measured wavelengths and propagation angles in Fig. 2(f). For each wave numbered 1 to 8, blue dots show the input wavelength $\lambda_{IN}$ against measured wavelength $\lambda_{OUT}$, indicating that the 2DST measures the horizontal wavelengths and propagation angles in the test image very well. Generally, shorter wavelengths are well-resolved but longer wavelengths are slightly underestimated. This  may be due to the

30 coarse spectral resolution of DFT-based methods for waves with wavelengths that are a large fraction of the image size, since such waves  can be more susceptible to spectral leakage problems.

**4  An alternative spectral window**

The use of the Gaussian window in the S-transform has some convenient mathematical advantages; it is analytically simple and has a definite integral over an infinite range. However, when it is used for 2-D S-transform analysis an unfortunate side effect  of the Gaussian window is the poor recovery of wave amplitude, discussed in the previous section.

5  Although a Gaussian is traditionally used, any suitable apodizing function may be used,  so long as its spatial integral is equal to unity (Stockwell, 2007). For example, Pinnegar and Mansinha (2003) used an asymmetric hyperbolic time-domain window for enhanced measurement of the onset times of one-dimensional time series components.

In this section, we  introduce a new spectral windowing function for the 2DST.

10   This new function takes the shape of an ellipse in the wavenumber domain, and a first-order Bessel function of the first kind $\mathbf{J}_1(z)$ function with a scaled $1/z$ envelope in the spatial domain

(for definition of $z$ see Eqn. 14 below). For this reason we refer to this window as the Elliptic-Bessel (E-B) window. We

15  find that  when AIRS measurements are analysed with the 2DST using this new Elliptic-Bessel window in place of the traditional Gaussian, the measurement of gravity wave amplitudes is greatly improved. Spectral resolution is also improved slightly, without adversely compromising spatial resolution.

**4.1  The Elliptic-Bessel window**

20  As discussed in Sect. 3.1, the spectral peaks in a DFT spectrum have a characteristic width, where the spectral power is spread in a broad peak around the central wavenumber. This spectral power is slightly reduced when a Gaussian window is applied, due to the immediate decrease in the Gaussian function around the central location.  The effect can be mitigated, but not fully reconciled, by decreasing the scaling parameter $c$, which broadens the  Gaussian window in the wavenumber domain.

25  However, this decreases the width of the spatial window, which increases the effect of wave undersampling for low wavenumbers.

One solution to this problem is to as

$$\omega_{ell}(\alpha_x, \alpha_y, k_x, k_y) = \begin{cases} 0 & \text{for} & 2\pi c \sqrt{\frac{(\alpha_x - k_x)^2}{k_x^2} + \frac{(\alpha_y - k_y)^2}{k_y^2}} & \geqslant & 1 \\ 1 & \text{for} & 2\pi c \sqrt{\frac{(\alpha_x - k_x)^2}{k_x^2} + \frac{(\alpha_y - k_y)^2}{k_y^2}} & < & 1 \end{cases} \quad .$$

 use a window that is an ellipse in the wavenumber domain. Here we introduce an Elliptic-Bessel window $W_{eb}$, defined in the wavenumber domain as the ellipse

$$W_{eb}(\alpha_x, \alpha_y, k_x, k_y) = \begin{cases} 0 \text{ for } \left(\frac{\alpha_x}{a}\right)^2 + \left(\frac{\alpha_y}{b}\right)^2 \geqslant 1 \\ 1 \text{ for } \left(\frac{\alpha_x}{a}\right)^2 + \left(\frac{\alpha_y}{b}\right)^2 < 1 \end{cases} \tag{12}$$

where $a = |k_x|/2\pi c$ and $b = |k_y|/2\pi c$ are the widths in the $\alpha_x$ and $\alpha_y$ directions . We see that the semi-major and semi-minor axes of this "voice ellipse" scale with angular wavenumbers and are equal to the standard deviations of the equivalent voice Gaussian window in  Eqn. 11.

A key feature of this new window is that, in the wavenumber domain, it does not immediately decrease with displacement from the central location, but rather has a scalable elliptical region within which the function is equal to unity. Thus, the window captures a much greater extent of the targeted spectral peak at $k_x$ and $k_y$,  which can greatly improve wave amplitude recovery compared to the traditional Gaussian window. The width of  $W_{eb}$ can also be more carefully adjusted using the scaling parameter $c$. This  window can then be used in place of the Gaussian windowing term in  Eqn. 10.

 One requirement for any apodizing window used in the Stockwell transform is that the spatial integral of the function must be equal to unity. This is so that the spatial integral of the Stockwell transform is equal to the Fourier transform $H(k_x, k_y)$ (Mansinha et al., 1997a), namely

$$\int_{-\infty}^{\infty} \int_{-\infty}^{\infty} S(\tau_x, \tau_y, k_x, k_y) \, d\tau_x \, d\tau_y = H(k_x, k_y) \tag{13}$$

which has the useful result of making the 2DST  fully invertible (Stockwell, 1999, 2007).

The normalisation term $|k_x||k_y|/8\pi^3 c^2$ in the traditionally-used Gaussian window in Eqn. 9 ensures that the Gaussian window satisfies this requirement. To check that the Elliptic-Bessel function is admissible as an apodizing function, we must first find its spatial form, then check that its spatial integral is also equal to unity.

The Elliptic-Bessel window $W_{eb}(\alpha_x, \alpha_y, k_x, k_y)$ is easily defined in the wavenumber domain as an ellipse, but its spatial form, which we denote as $w_{eb}(x, y, k_x, k_y)$ is given as

$$w_{eb}(x, y, k_x, k_y) = \frac{|k_x||k_y|}{8\pi^3 c^2} \frac{\mathbf{J}_1(z)}{z} \tag{14}$$

where $\mathbf{J}_1$ is the first-order Bessel function of the first kind (Abramowitz and Stegun, 1964) and

$$z = \frac{1}{2\pi c} \sqrt{k_x^2 x^2 + k_y^2 y^2}.$$

A short derivation of the function in Eqn. 14 is provided in Appendix A1. Fortunately, the spatial integral of Eqn. 14 is indeed equal to unity, proof of which is presented in Appendix A2. This confirms that the Elliptic-Bessel window is admissible as an apodizing function in the 2-D Stockwell transform, and validates its use in this study and beyond.

To recap our notation in this study, we have described two windowing functions for the

5  2-D Stockwell transform: the traditional Gaussian and the new Elliptic-Bessel windows, which we denote in the spatial domain as $w_g(x, y, k_x, k_y)$ and $w_{eb}(x, y, k_x, k_y)$ respectively, and in the wavenumber domain as $W_g(\alpha_x, \alpha_y, k_x, k_y)$ and $W_{eb}(\alpha_x, \alpha_y, k_x, k_y)$ respectively, where the $W_g$ and $W_{eb}$ are the Fourier transforms of $w_g$ and $w_{eb}$.

**4.2**

10 Figure 3 shows three-dimensional surface plots of the spatial and wavenumber domain forms of the traditional 2-D Gaussian and new Elliptic-Bessel windows used in the 2-D Stockwell transform here.

 The surfaces in Figures 3(a) and 3(c) show the wavenumber-domain and spatial-domain forms of the Gaussian window, for arbitrary wavenumbers $k_x$ and $k_y$. As discussed above, the maximum value of the Gaussian is equal to unity in the

15 wavenumber domain, but equal to $|k_x||k_y|/8\pi^3 c^2$ in the spatial domain such that its spatial integral is equal to unity. This is a requirement of any windowing function in the S-transform. The standard deviations of the $w_g$ and $W_g$ scale with wavenumbers $k_x$ and $k_y$ as described in Sect. 2.2 (Eqns. 9 and 11), providing the voice Gaussian.

Likewise, Figures 3(b) and 3(d) show the wavenumber-domain ($W_{eb}$) and spatial-domain ($w_{eb}$) forms of the Elliptic-Bessel window, for the same arbitrary wavenumbers $k_x$ and $k_y$ as used for the Gaussian windows in panels (a) and (c). The semi-major

20 and semi-minor axes of the elliptic region in 3(b) are scaled with wavenumbers $k_x$ and $k_y$, and are equal to the standard deviations of the equivalent Gaussian in 3(a), providing the voice ellipse.

The spatial-domain form of the Elliptic-Bessel window $w_{eb}$, described by a Bessel-shaped function within an envelope, is shown in Fig.  3(d) and described by Eqn. 14. The maximum value of $w_{eb}$ is $|k_x||k_y|/16\pi^3 c^2$, which ensures that its spatial integral is equal to unity. This is equal to half of the maximum value of the equivalent Gaussian

25 in Fig. 3(c), since the terms involving $z$ in Eqn. 14 tend to $1/2$ as

$$\omega_{sinc}(\alpha_x, \alpha_y, k_x, k_y) = \frac{\sin\left(2\pi c\sqrt{R_{\alpha_x}^2/k_x^2 + R_{\alpha_y}^2/k_y^2}\right)}{2\pi c\sqrt{R_{\alpha_x}^2/k_x^2 + R_{\alpha_y}^2/k_y^2}}$$

 $x \to 0$ and $y \to 0$. The width of the central region

30 of $w_{eb}$ is very slightly wider than the equivalent Gaussian, resulting in slightly coarser spatial resolution. However, the ability to "tune" the 2DST with the scaling parameter $c$ ensures that this effect can be compensated by a reasonable trade-off.

In the next section we show that the use of the Elliptic-Bessel window  in the  2DST, in place of the traditional Gaussian window, significantly improves wave amplitude recovery. This is very useful for our analysis of AIRS data in Sect. 5.

**4.2 Invertibility**

A very convenient aspect of the S-transform is its invertibility. Since we have shown here that both the traditional Gaussian and new Elliptic-Bessel windows have spatial integrals equal to unity, the 2DST can be completely inverted to recover the original 2-D image, whichever of these windows or real non-zero positive values of the scaling parameter $c$  are used. Note that a traditional 1-D or 2-D CWT does not necessarily have this capability. The fact that we can achieve such flexibility in spatial-spectral resolutions by swapping windows or by adjusting $c$, yet still retain the capability of inversion, further highlights the strength of the 2DST as a tool for spatial-spectral analysis of geophysical data.

Unfortunately, to take full advantage of DFT algorithms and the inversion capability of the 2DST for AIRS data, we must compute the 2DST using all permitted wavenumber voices in both dimensions. This requires nearly 12 000 inverse DFT calculations for each AIRS granule using the traditional voice-by-voice implementation described here, the computational load of which could be quite impractical for large-scale studies. Interpolating AIRS measurements to a coarser resolution with fewer pixels could be one solution to reduce computational cost, but this will obviously undersample short horizontal wavelengths in the data. Faster methods for computing the S-transform have been developed (Brown et al., 2010) which may increase practicality in the future. Other steps, such as avoiding programming loops and ensuring that any 2-D objects to be transformed have dimensions that are powers of two, may also reduce relative computational expense.

**4.3 The effect of window choices on AIRS granules**

Figure 4 shows an AIRS granule over the Southern Andes measured on 24[th] May 2008, analysed using the 2DST with three different windowing approaches.

In Figs. 4(b) and 4(e), we use a Gaussian windowing function with the scaling parameter $c$  set to one. This is the window usually used in 1DST implementations. We see that, as discussed above, this choice of window is only able to recover the very general, long-horizontal wavelength features of the granule, with poor spatial localisation and significantly reduced amplitude. This is due to a large proportion of the spectral response being lost by the windowing Gaussian when applied to two dimensions.

We can reduce the impact of this by decreasing the scaling parameter $c$, which broadens (narrows) the spectral (spatial) window. This provides improved amplitude recovery and improved spatial localisation at the expense of spectral localisation (Fritts et al., 1998). Since we only select a single dominant spectral peak for each location on the granule, this is acceptable for our

purposes. The "reconstructed" perturbations and horizontal wavelengths (Figs. 4(c) and (f)) are now much more representative of the wave features in the granule.

One problem remains, however. By decreasing $c$, we narrow our spatial window. In regions where wave amplitudes are low, such as the bottom-left corner of Fig. 4(a), this narrow Gaussian window starts to undersample long wavelengths, such that only very short wavelengths are attributed to the region. The  Elliptic-Bessel window used in Figs. 4(d) and (g) performs better at recovering the underlying larger-scale structure of the granule, without defaulting to the small-scale noisy variations. Amplitude recovery at all wavelengths is also improved over either of the Gaussian approaches.

In the general case, these low-amplitude, small-scale variations are unlikely to be due to gravity waves with vertical wavelengths visible to AIRS, so their recovery is something we try to avoid. Furthermore, such wavelengths are very close to or at the Nyquist limit for these data. Our confidence in their measurement is thus very low, yet the momentum fluxes they transport can dominate. We discuss this further in Sect. 5.4.

For the windowing functions considered, it is clear from Fig. 4 that the scaling parameter $c$ has a  significant effect in determining the spatial-spectral localisation capabilities of the 2DST. The  Elliptic-Bessel windowing function, with a scaling parameter of $c = 0.25$, was selected for our AIRS analysis in the next section. This choice provided the best trade-off between spatial and spectral localisation of different wave regimes in AIRS measurements.

As discussed in Sect. 3.1, the 2-D images $\xi(\tau_x, \tau_y)$, $K_x(\tau_x, \tau_y)$ and $K_y(\tau_x, \tau_y)$ contain the dominant measured wave amplitudes and wavelengths at each location on the granule. These images are computed on a pixel-by-pixel basis, selecting a single monochromatic wave with the largest amplitude in the localised spectrum for each pixel.

As a result, the reconstructed images shown in Fig. 4 (b-d), computed by taking the real part of the complex image $\xi(\tau_x, \tau_y)$, will never be perfect representations of the input data, but provide a "best guess" of the dominant features of the granule.

**4.4**

Since we have shown that the 2DST is fully invertible for both the Gaussian and Elliptic-Bessel windowing approaches (Sections 4 and Appendix A), a complete reconstruction of the input image is of course producible by taking the "inverse" of the full four-dimensional 2DST object, but here we desire 2-D "maps" of wave properties, so a best guess method is used.

$$\int_{-\infty}^{\infty} S(\tau, f)\, d\tau = H(f),$$

which can be easily inverted to recover the original signal. This feature of the S-transform is dependent on the requirement that the temporal (or spatial) sum of the selected windowing function $\omega$ is equal to unity, namely

$$\int\limits_{-\infty}^{\infty} \omega(\tau - t, f)\, d\tau = 1, \quad \text{or} \quad \int\limits_{-\infty}^{\infty} \omega(\tau_x - x, k_x)\, dx = 1.$$

By taking the spatial sum of the Gaussian window in Eq. 4, we see that the normalisation term $\frac{|k_x||k_y|}{2\pi c^2}$ ensures that this condition is satisfied. The spatial domain form of our spectral-domain elliptical windowing function $\omega_{ell}(\alpha_x, \alpha_y, k_x, k_y)$ in Eq. ?? is sinc-shaped and given by

$$\omega_{ell}(\tau_x, \tau_y, k_x, k_y) = K_{ell} \times \text{sinc}\left(\frac{1}{2\pi c}\sqrt{(x - \tau_x)^2 k_x^2 + (y - \tau_y)^2 k_y^2}\right)$$

where $K_{ell}$ is a normalisation factor and the sinc function is the unnormalised form (no factor of $\pi$). Unfortunately, the spatial integral of A possible quantitative metric to assess the first-order effectiveness of our 2DST analysis in Fig. 4 could be to compare the variance of the sinc term in ?? does not have a definite value, which means we cannot obtain an exact expression for $K_{ell}$ input image with the variances of each of the reconstructions. However, a good approximation can be made if we take $K_{ell} = \frac{|k_x||k_y|}{4\pi c^2}$. This is equal to one half of the normalisation term used for the Gaussian window. Using this approximation, we are able to invert the 2DST when our elliptical windowing function is used, although this approximation can lead to under-representation of very low wavenumbers.

This means that, if required, the 2DST can be exactly inverted for the Gaussian and pseudo-inverted for the elliptical windowing functions presented here. Thus, since the reconstructions are computed as a best guess method on a pixel-by-pixel basis, their total variance is not readily related to the total variance of the input image and thus may not be meaningful as a comparison. Furthermore, such use of the original 2-D image can be recovered from the inverse DFT of the spatial sum of the 2DST. Note that a traditional CWT does not have this capability. The fact that we can achieve such flexibility in spatial-spectral resolutions by swapping our Gaussian window for an elliptical or by adjusting $c$, yet still retain the capability of inversion, further highlights the strength of the 2DST as a tool for spatial-spectral analysis of geophysical data. image variance would only be appropriate if the distribution of perturbations was unimodal and ideally Gaussian, which is not the case for an image of a sinusoidal wave. In practice however, we generally expect the variance of the reconstruction not to exceed the variance of the input image, since wave amplitudes computed on a pixel-by-pixel basis from a localised spectrum will usually be underestimated for the reasons given in Sect. 3.1.

Unfortunately, to take full advantage of DFT algorithms and the inversion capability of It is not impossible that in some rare cases the total variance of the 2DST for AIRS data, we must compute the 2DST using all permitted wavenumber voices in both dimensions. This requires nearly 12 000 inverse DFT calculations for each AIRS granule using the traditional voice-by-voice implementation described here. Interpolating AIRS measurements to a coarser resolution with fewer pixels could be one solution, but this will obviously undersample short horizontal wavelengths in the data. Faster methods for computing the S-transform have been developed (Brown et al., 2010) which may increase practicality in reconstruction could exceed the total

variance of the input image, for example due to the  spatial extent of a wave feature being slightly over-estimated. If the localised spectrum for one pixel is affected a larger amplitude wave feature in one of its neighbouring pixels, this can result in subtle artificial "borders" between different wave regimes in the reconstructions. This is not a limitation of the 2DST itself, but arises in the somewhat forced extraction of localised gravity-wave parameters contained in the 4-D Stockwell transform object $S(\tau_x, \tau_y, k_x, k_y)$ in order to produce the 2-D image. This effect should be carefully considered in future work to ensure wave properties are not over-represented.

**5 AIRS gravity wave analysis using the 2DST**

[revised manuscript text omitted]

Generally, the agreement between reconstructed wave features in Figure 5(b) and AIRS measurements in Figures 5(a) is very good, but some uncertainties remain. As discussed in Sect. 4.3, there is some discrepancy regarding the spatial extent of some wave features, such as a small positive wave crest located just south west of South Georgia which appears to be located slightly

east, with an apparently slightly over-estimated amplitude, than is observed in the AIRS measurements. Conversely, at 63°S 28°W, a positive bow-shaped wave crest is observed in the AIRS measurements but is under-estimated in the reconstruction. As mentioned in Sect. 4.3, these small misrepresentations are not a limitation of the 2DST itself, but rather the forced extraction of gravity wave parameters from the 4-D S-transform object in order to create the 2-D reconstruction, where only one single wave feature with the largest localised spectral amplitude is assigned at each location. The overall agreement is still very good, but future work to improve the extraction of gravity wave parameters from the 4-D S-transform object may help to resolve some of these discrepancies.

Panels 5(c) and 6(c) show  full underlying wave amplitudes $|T'|_{2DST}$ for each granule. This is found by taking the absolute magnitude of the complex 2DST object $\xi(\tau_x, \tau_y)$ as described in Sect. 3.1. This property provides us with a useful metric with which to define regions of the granule which do or do not contain wave-like perturbations, such that we can limit spurious detections (e.g. McDonald, 2012). In Figs. 5(h) and 6(c–h), we exclude regions of each granule where the  underlying wave amplitude is more than one standard deviation below the mean  underlying wave amplitude of the granule. In Figs. 5(c–g), we do not exclude such regions for discussion purposes, so as to provide an example of the data we would otherwise omit.

Panels 5(d) and 6(d) show absolute horizontal wavelengths $\lambda_H = (k_{AT}^2 + k_{XT}^2)^{-1/2}$, where $k_{AT}$ and $k_{XT}$ are the along-track and cross-track wavenumbers respectively. We can see that these horizontal wavelengths clearly define different regimes of the dominant wave features of the granules, as in the test case in Sect. 3.1, though the AIRS data are more complex. In the South Georgia granule in Fig. 5(d), we see that the island lies within a wave field where long horizontal wavelengths are dominant around and to the east of the island over the ocean, with their wavenumber vectors aligned roughly parallel to the direction of the mean flow. This is characteristic of a wing-shaped mountain wave field (Alexander and Grimsdell, 2013), and is in good agreement with visual inspection of the granule itself.

In panels 5(e) and 6(e), we show the orientation of the horizontal wavenumber vector measured anticlockwise from east. $\theta$ is calculated by first projecting the along-track and cross-track wavenumber vectors $k_x$ and $k_y$ into their zonal and meridional components $k$ and $l$ using the azimuths of the along-track and cross-track directions at each location on the granule, then taking $\theta = \tan^{-1}(\frac{l}{k})$. Note that $\theta$ only describes the orientation and not the true horizontal direction of propagation of the wavenumber vectors, which retain a $\pm 180°$ ambiguity that we break below.

In the South Georgia granule (Fig. 5), we see that our 2DST measurements in the southern  region of the granule are largely dominated by small-scale, low-amplitude, short horizontal wavelength features with random directions of propagation. Most of these features are likely to be due to noise and not attributable to coherent wave structures. By using a threshold amplitude, such regions are effectively removed, leaving well-defined regions with clear wavelike perturbations. The contribution of small-scale features  that remain after this step is discussed further in Sect. 5.4.

**5.3 Momentum fluxes**

Here we make estimates of gravity wave momentum flux for the dominant wave-like features measured by the 2DST in our selected granules, following the method of Alexander et al. (2009).

 Ern et al. (2004) showed that the zonal and meridional components of gravity wave momentum flux $MF_x$ and $MF_y$  can be given by

$$\left(MF_{x,y}x, MF_y\right) = \frac{\rho}{2}\left(\frac{g}{N}\right)^2\left(\frac{T'_a}{\bar{T}}\right)^2\left(\frac{k}{m}, \frac{l}{m}\right) \tag{15}$$

where $\rho$ is density at a height of 40 km, $g$ is the acceleration due to gravity, $N$ is the buoyancy frequency, $T'_a$ is the attenuation-scaled  full underlying wave amplitude, $\bar{T}$ is the background temperature, and $k$, $l$ and $m$ are wavenumbers in the zonal, meridional and vertical directions respectively  Eqn. 15 assumes the mid-frequency approximation (Fritts and Alexander, 2003, their Sect. 2.1), which is valid for a large portion of the observable gravity wave spectrum. For these waves, $m^2 \ll (k^2 + l^2)$ and Coriolis effects are neglible, as is the case for nearly all waves measured here due to the size of the AIRS beam footprint (Hoffmann et al., 2014) and the vertical weighting function of the 667.77 cm$^{-1}$ channel (Alexander and Barnet, 2007). When compared with momentum fluxes calculated without making the mid-frequency approximation, Ern et al. (2004) found discrepancies typically not exceeding around 10%.

[revised manuscript text omitted]

To conclude, our new 2DST-based gravity wave analysis method for AIRS data makes significant improvements over current methods in several key areas, and we would advocate its use in future work.

**Appendix A: Admissibility of the Elliptic-Bessel window in the Stockwell Transform**

In Sect. 4 we introduced the Elliptic-Bessel window as new apodizing function for the 2-D Stockwell transform (2DST). One requirement for any apodizing function for use in the Stockwell transform is that its spatial sum must be equal to unity. If this condition is satisfied, the spatial sum of the 2DST is equal to the 2-D Fourier transform, making the 2DST fully invertible.

5    In this appendix we demonstrate that the Elliptic-Bessel window is admissible as an apodizing function in the S-transform. To do this, we must first find the spatial analogue of the wavenumber-domain ellipse we defined in Eqn. 12. We must then take the spatial integral of this function to demonstrate that it is equal to unity.

**A1   The Elliptic-Bessel window in the spatial domain**

The Elliptic-Bessel window is defined in the wavenumber $(\alpha_x, \alpha_y)$ domain as

10    $$W_{eb}(\alpha_x, \alpha_y, k_x, k_y) = \begin{cases} 0 \text{ for } \left(\frac{\alpha_x}{a}\right)^2 + \left(\frac{\alpha_y}{b}\right)^2 \geqslant 1 \\ 1 \text{ for } \left(\frac{\alpha_x}{a}\right)^2 + \left(\frac{\alpha_y}{b}\right)^2 < 1 \end{cases} \tag{A1}$$

where $a = |k_x|/2\pi c$ and $b = |k_y|/2\pi c$ are the half-widths of the ellipse in the $\alpha_x$ and $\alpha_y$ directions (see Fig. 3(b)). The spatial-domain form of the Elliptic-Bessel window, denoted here by $w_{eb}(x, y, k_x, k_y)$, is found by taking the inverse 2-D Fourier transform of Eqn. A1 as

$$w_{eb}(x, y, k_x, k_y) = \mathscr{F}_x^{-1} \mathscr{F}_y^{-1} \left[ W_{eb}(\alpha_x, \alpha_y, k_x, k_y) \right] = \frac{1}{4\pi^2} \int\limits_{\infty}^{\infty} \int\limits_{\infty}^{\infty} W_{eb}(\alpha_x, \alpha_y, k_x, k_y) e^{i(\alpha_x x + \alpha_y y)} d\alpha_x \, d\alpha_y \tag{A2}$$

15    Since $W_{eb}(\alpha_x, \alpha_y, k_x, k_y) = 1$ within the ellipse and zero everywhere else, and has double symmetry, we can change the limits of integration to be the boundaries of the ellipse, expressing the total integral as a sum of four equal quadrants

$$w_{eb}(x, y, k_x, k_y) = \frac{4}{4\pi^2} \int\limits_{0}^{\sqrt{b^2 - \frac{\alpha_x^2 b^2}{a^2}}} \int\limits_{0}^{a} e^{i(\alpha_x x + \alpha_y y)} d\alpha_x \, d\alpha_y \tag{A3}$$

We then recognise that the exponential term in the transform above can be replaced with sine and cosine functions as

$$e^{i(\alpha_x x + \alpha_y y)} = \left( \cos(\alpha_x x) + i \sin(\alpha_x x) \right) \left( \cos(\alpha_y y) + i \sin(\alpha_y y) \right)$$

20    $$= \cos(\alpha_x x) \cos(\alpha_y y) + i \sin(\alpha_x x) \cos(\alpha_y y) + i \sin(\alpha_y y) \cos(\alpha_x x) - \sin(\alpha_x x) \sin(\alpha_y y) \tag{A4}$$

We can omit the last three terms in A4 since, due to the symmetry of the sine function around $(0, 0)$, each term will eventually sum to zero. We can then rewrite Eqn. A3 as

$$w_{eb}(x, y, k_x, k_y) = \frac{4}{4\pi^2} \int\limits_{0}^{\sqrt{b^2 - \frac{\alpha_x^2 b^2}{a^2}}} \int\limits_{0}^{a} \cos(\alpha_x x) \cos(\alpha_y y) d\alpha_x \, d\alpha_y \tag{A5}$$

This integral can be further simplified if we switch to polar coordinates using the substitutions $\alpha_x = ar\cos(\phi)$ and $\alpha_y = br\sin(\phi)$ after which the expression in A5 becomes

$$w_{eb}(x,y,k_x,k_y) = \frac{ab}{4\pi^2} \int\limits_0^{2\pi} \int\limits_0^1 \cos(arx\cos(\phi))\cos(bry\sin(\phi))\, r\, dr\, d\phi \tag{A6}$$

Next we substitute $\mathcal{A} = arx$ and $\mathcal{B} = bry$ and, using multiple angle formulae, rewrite A6 as

$$w_{eb}(x,y,k_x,k_y) = \frac{ab}{4\pi^2} \int\limits_0^1 r \int\limits_0^{2\pi} \cos(\mathcal{A}\cos(\phi))\cos(\mathcal{B}\sin(\phi))\, d\phi\, dr$$

$$= \frac{ab}{8\pi^2} \int\limits_0^1 r \int\limits_0^{2\pi} [\cos(\mathcal{A}\cos\phi + \mathcal{B}\sin\phi) + \cos(\mathcal{A}\cos\phi - \mathcal{B}\sin\phi)]\, d\phi\, dr \tag{A7}$$

$$= \frac{ab}{8\pi^2} \int\limits_0^1 r \int\limits_0^{2\pi} [\cos(\sqrt{\mathcal{A}^2 + \mathcal{B}^2}\cos(\phi - \Lambda)) + \cos(\sqrt{\mathcal{A}^2 + \mathcal{B}^2}\cos(\phi + \Lambda))]\, d\phi\, dr \tag{A8}$$

where $\Lambda = \tan^{-1}(\mathcal{B}/\mathcal{A})$. Here, $\Lambda$ is simply an arbitrary phase due to the periodicity of the cosine function when integrated over 0 to $2\pi$, so the integrals of both terms in the square brackets in Eqn. A8 will be equal. Hence we can simply add these terms such that we have

$$w_{eb}(x,y,k_x,k_y) = \frac{ab}{8\pi^2} \int\limits_0^1 r \int\limits_0^{2\pi} 2\ \cos\left(\sqrt{\mathcal{A}^2 + \mathcal{B}^2}\cos\phi\right)\, d\phi\, dr \tag{A9}$$

Next we recall the integral definition of the zeroth-order Bessel function of the first kind $\mathbf{J}_0(x)$ (Abramowitz and Stegun, 1964) given as

$$\mathbf{J}_0(x) = \frac{1}{2\pi} \int\limits_0^{2\pi} \cos(x\cos\phi)\, d\phi \tag{A10}$$

and substitute into Eqn. A9 and reintroduce our substitutions of $\mathcal{A} = arx$ and $\mathcal{B} = bry$ to give

$$w_{eb}(x,y,k_x,k_y) = \frac{ab}{2\pi} \int\limits_0^1 \mathbf{J}_0\left(r\sqrt{a^2x^2 + b^2y^2}\right) r\, dr \tag{A11}$$

We now use a new substitution that $\xi = r\sqrt{a^2x^2 + b^2y^2}$ and rewrite Eqn. A11 as

$$w_{eb}(x,y,k_x,k_y) = \frac{ab}{2\pi} \int\limits_0^{\sqrt{a^2x^2 + b^2y^2}} \frac{\xi\, \mathbf{J}_0(\xi)}{a^2x^2 + b^2y^2}\, d\xi \tag{A12}$$

Next we use the standard result (e.g. Abramowitz and Stegun, 1964) that

$$\int_{x_1}^{x_2} x\, \mathbf{J}_0(x)\; dx \;=\; x\, \mathbf{J}_1(x)\, \Big|_{x_1}^{x_2} \tag{A13}$$

to rewrite Eqn. A12 as

$$w_{eb}(x,y,k_x,k_y) = \frac{ab}{2\pi}\frac{\xi\, \mathbf{J}_1(\xi)}{(a^2x^2+b^2y^2)}\,\Bigg|_0^{\sqrt{a^2x^2+b^2y^2}}$$

$$= \frac{ab}{2\pi}\frac{\mathbf{J}_1\left(\sqrt{a^2x^2+b^2y^2}\right)}{\sqrt{a^2x^2+b^2y^2}} \tag{A14}$$

Finally, recalling that $a = |k_x|/2\pi c$ and $b = |k_y|/2\pi c$ are the half-widths of the original ellipse in A1, we now can write the analytical expression for the spatial form of the Elliptic-Bessel window as

$$w_{eb}(x,y,k_x,k_y) \;=\; \frac{|k_x||k_y|}{8\pi^3 c^2}\frac{\mathbf{J}_1(z)}{z} \tag{A15}$$

where

$$z \;=\; \frac{1}{2\pi c}\sqrt{k_x^2 x^2 + k_y^2 y^2} \tag{A16}$$

This spatial-domain form of the Elliptic-Bessel window in Eqn. A15 is plotted in Fig. 3(d).

Equation A15 describes a $\mathbf{J}_1(z)$ function within a scaled $1/z$ envelope. Because of this, the terms involving $z$ in A15 converge to $1/2$ as $x \to 0$ and $y \to 0$, such that the central region of the function has peak value of $|k_x||k_y|/16\pi^3 c^2$, as shown in Fig. 3(d). Interestingly, this value is equal to half the peak value of the equivalent Gaussian window shown in Fig. 3(c). The central peak of the Elliptic-Bessel window is also, for each frequency voice, slightly broader than that of the equivalent voice Gaussian.

**A2   Spatial integral of the Elliptic-Bessel window**

Now that we have found an analytical expression for the spatial-domain form of the Elliptic-Bessel window (Eqn. A15), we can proceed to check that it is admissible as an apodizing function in the 2-D Stockwell transform; namely that its spatial sum is equal to unity (e.g. Pinnegar and Mansinha, 2003). The spatial sum of Eqn. A15, denoted here by $\mathbb{I}$, can be written as

$$\mathbb{I} = \int_{-\infty}^{\infty}\int_{-\infty}^{\infty} \frac{|k_x||k_y|}{8\pi^3 c^2}\frac{\mathbf{J}_1(z)}{z}\; dx\, dy \tag{A16}$$

This integral can be simplified if we reintroduce our substitutions $a = |k_x| / 2\pi c$ and $b = |k_y| / 2\pi c$ and switch to polar coordinates, using the substitutions $x = \frac{\Lambda \cos(\varphi)}{a}$ and $y = \frac{\Lambda \sin(\varphi)}{b}$ to give

$$\mathbb{I} = \frac{ab}{2\pi} \int\limits_0^{2\pi} \int\limits_0^{\infty} \frac{\mathbf{J}_1(\Lambda)}{\Lambda} \frac{\Lambda}{ab} \; d\Lambda \, d\varphi$$

$$= \frac{1}{2\pi} \int\limits_0^{2\pi} \int\limits_0^{\infty} \mathbf{J}_1(\Lambda) \; d\Lambda \, d\varphi$$

$$= \frac{2\pi}{2\pi} \int\limits_0^{\infty} \mathbf{J}_1(\Lambda) \; d\Lambda \tag{A17}$$

Using the standard result (e.g. Abramowitz and Stegun, 1964) that

$$\int\limits_{x_1}^{x_2} \mathbf{J}_1(x) \; dx \;\; = \;\; -\mathbf{J}_0(x) \, \Big|_{x_1}^{x_2} \tag{A18}$$

we see that Eqn. A17 becomes

$$\mathbb{I} = -\mathbf{J}_0(\Lambda) \, \Big|_0^{\infty}$$

$$= (0) - (-1)$$

$$= 1 \tag{A19}$$

as required. This result confirms that the spatial sum of $w_{eb}(x, y, k_x, k_y)$ is indeed equal to unity, thus the Elliptic-Bessel window is admissible as an apodizing window for the 2-D Stockwell transform.

**A3 Admissibility of other windows**

In this appendix so far, we have found a useful analytical expression for spatial form of the Elliptic-Bessel window presented in this study. We have then shown that its spatial integral is equal to unity and it is thus admissible as an apodizing function in the 2-D Stockwell Transform. In other cases, a quick test may be performed on candidate S-transform windowing functions to check if this spatial integral is unity.

If we take the spatial integral $\mathbb{I}$ of the spatial-domain form of a candidate windowing function $w(x, y, k_x, k_y)$, namely

$$\mathbb{I} = \int\limits_{-\infty}^{\infty} \int\limits_{-\infty}^{\infty} w(x, y, k_x, k_y) \, dx \, dy \tag{A20}$$

and introduce the factor $e^{-i(k_x x + k_y y)}$, noting that when $k_x = k_y = 0$ this factor is equal to unity, then $\mathbb{I}$ can be written as

$$\mathbb{I} = \int_{-\infty}^{\infty} \int_{-\infty}^{\infty} w(x, y, k_x, k_y) e^{-i(k_x x + k_y y)} \, dx \, dy \bigg|_{k_x = k_y = 0} \tag{A21}$$

$$= W(\alpha_x, \alpha_y, k_x, k_y) \bigg|_{k_x = k_y = 0} \tag{A22}$$

where $W(\alpha_x, \alpha_y, k_x, k_y)$ is the wavenumber domain form of the candidate window and the notation $|_{k_x = k_y = 0}$ denotes that the function is evaluated at $k_x = k_y = 0$. This means that if the value of $W(\alpha_x, \alpha_y, k_x, k_y)$ evaluated at $k_x = 0$ and $k_y = 0$ is equal to unity, then its spatial integral will also be equal to unity. If it is not, then the candidate window is not admissible for use the Stockwell transform. Figures 3(a) and 3(b) show that both the Gaussian window and the Elliptic-Bessel window are equal to unity at $k_x = k_y = 0$, and thus satisfy this requirement. This short test may be helpful in the design of proposed alternative S-transform windowing functions in the future.

*Acknowledgements.* NPH is funded by a NERC studentship awarded to the University of Bath. CJW and NJM are supported by NERC grant NE/K015117/1. The authors would like to thank the AIRS programme team for many years of hard work producing the data used here, and also the anonymous reviewers for their helpful suggestions.

**References**

Abramowitz, M. and Stegun, I.: Handbook of Mathematical Functions: With Formulas, Graphs, and Mathematical Tables, Applied mathematics series, Dover Publications, 1964.

[revised manuscript text omitted]